# TreeTop: Topology-Aware Fine-Tuning for LLM Conversation Tree Understanding

## Abstract

While Large Language Models (LLMs) have dominated a wide diversity of natural language tasks, improving their capabilities on *structured* inputs such as graphs remains an open challenge. We introduce TreeTop, a fine-tuning framework for LLMs that significantly improves their ability to reason over structural relationships in multi-party discussion *trees*, e.g. on social media platforms. TreeTop is a novel set of 17 tasks designed to test the ability of LLMs to selectively focus on the structure and/or content of conversation tree graphs. We find that LLMs fine-tuned on TreeTop significantly outperform all baseline models (including state-of-the-art GNNs) in multiple settings: generalizing to unseen TreeTop tasks, and performance on downstream social media inference tasks (e.g. controversy detection), including their challenging "early-detection" variants. TreeTop charts new ground toward LLMs with generalized understanding of structured inputs.

## 1 Introduction

Large Language Models (LLMs) have achieved state-of-the-art over an extremely vast landscape of tasks that can be cast as token sequence-to-sequence problems (Zhao et al., 2023b; Srivastava et al., 2022), partially through the combined effect of instruction fine-tuning (Wei et al., 2021) and scaling (Chung et al., 2024; Zhang et al., 2023a). A current open challenge for LLM capabilities is the handling of structured inputs (such as tables, e.g. Sui et al. (2024)), where the output depends strictly on tokens distributed throughout the input according to a certain pattern.

Recently, graph-structured inputs have emerged as one of the new frontiers in structured inputs for LLMs (Pan et al., 2024; Tsitsulin et al., 2024; Chen et al., 2024b; Ye et al., 2024). Graphs are flexible data structures containing objects (nodes) and relationships (edges) for which a wide variety of models have been crafted to complex systems from social to biological networks (Wu et al., 2020; Zhang et al., 2020). Enabling LLMs to perform tasks on graph-structured inputs has the potential to considerably expand their application scope. Recent works have focused on improving LLM performance on classical graph problems such as edge existence and counting (Fatemi et al., 2023; Perozzi et al., 2024; Wu et al., 2024; Wang et al., 2024). These LLMs, through appropriate training or prompting, have already been shown to outperform Graph Neural Networks (GNNs) (Chami et al., 2022) on graph learning tasks like node classification and link prediction (Ye et al., 2024).

In this paper, we continue this line of research by developing LLM capabilities for *conversation graph* inputs: graphs that encode online forum discussions by mapping replies to their parent comments, stemming from the root post. Often called conversation "trees", these graphs are ubiquitous due to the proliferation of online social platforms, and they are the input to many important learning tasks, such as misinformation detection, controversy detection, and trend prediction (Olteanu et al., 2019; Zeng & Tang, 2021; Ji et al., 2021; Hessel & Lee, 2019). Furthermore, conversation trees are directed, acyclic, and temporal, creating unique topological learning challenges, and distinguishing them from the usually undirected, cyclic, and static graphs found in standard GNN benchmarks (Hu et al., 2020a). These unique properties have established conversation trees as a distinct subfield, drawing approaches from GNNs (Xu et al., 2023) to LSTMs (Mukiri & Burra, 2023) and their hybrids (Patel et al., 2022).

Another distinguishing aspect of conversation trees is that the graph itself defines a flow through a multi-agent discussion. Each node is attributed with a natural-language utterance, an utterance

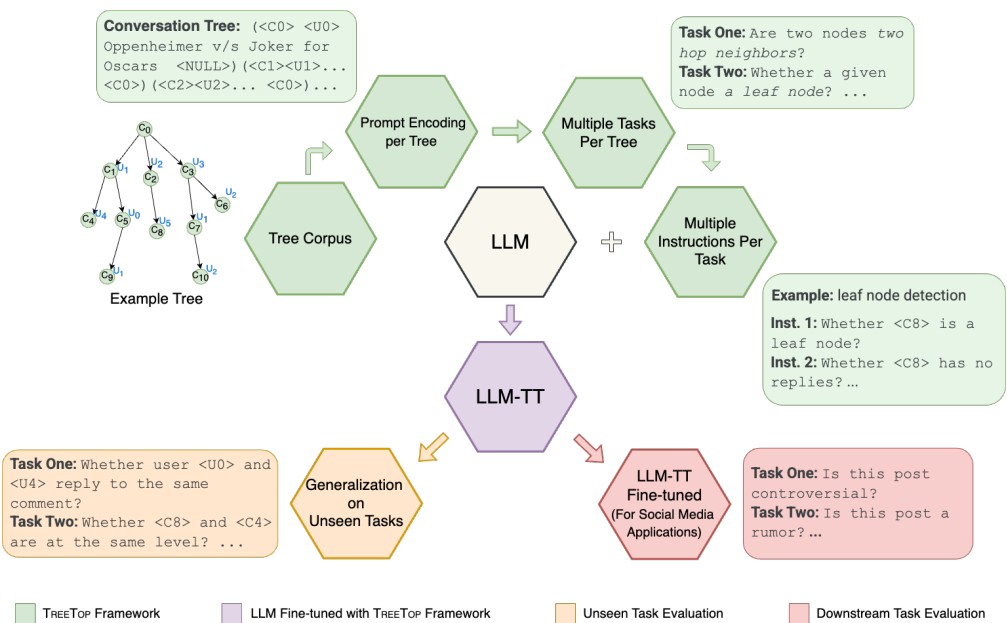

Figure 1: TREETOP framework (in ▢) provides a dataset of 12 structural tasks to improve the LLM's ability to understand and reason over structure via fine-tuning. The TREETOP framework also provides additional 5 unseen tasks to evaluate generalization of the LLM-TT model. Resultant models can be applied to multiple downstream conversation tree tasks.

which can be informed by any existing utterances in the conversation at the time of posting. Furthermore, the graph topology and the discussion can interact in potentially meaningful ways: a linear, back-and-forth sub-graph between two discussants has a signature quite different from the star-like pattern created, e.g., by a comment that receives many one-off replies. We observe that these phenomena create a potentially rich space of new challenges for LLMs at the intersection of structure and language. However, the application of LLMs to conversation trees has yet to receive attention. As a first step in this space, we introduce **Tree Top**ology-Aware Fine-Tuning (TREETOP): a novel learning framework for LLMs that significantly improves their performance across a range of tasks on conversation trees. As shown in Figure 1, TREETOP provides a collection of structural QA tasks defined over conversation trees that can be used to fine-tune an arbitrary LLM before zero/few-shot deployment or downstream fine-tuning. Specifically, our contributions are as follows:

**1.** We introduce novel machinery to enable LLMs to handle conversation trees, including a prompt framework and conversation tree encoding. We also make the code for the TREETOP framework available here[1] to advance further research in this field.

**2.** We propose TREETOP, a conversation graph fine-tuning framework including 17 structural tasks that target an LLM's ability to *reason* about the conversation, such as tree navigation and user-user reply counting. We show that fine-tuning using the TREETOP framework allows LLMs to generalize to even to *unseen* structural tasks.

**3.** Through extensive experimentation, we show that LLMs fine-tuned with TREETOP significantly outperform their regular counterparts, including state-of-the-art GNNs, on four social media tasks: controversial post detection (Hessel & Lee, 2019), rumor detection (Zubiaga et al., 2016), fake news detection (Nakamura et al., 2019), and winning argument thread detection (Tan et al., 2016). Our approach also demonstrates superior performance on "early detection" versions of these tasks.

## 2 RELATED WORK

**LLM-based approaches to graph problems.** Graph learning with LLMs is a nascent area of research (Tang et al., 2023; He et al., 2023). Guo et al. (2023) study whether LLMs can understand

---

[1]https://tinyurl.com/treetopframework

graph structural information, and Huang et al. (2023) show that LLMs tend to process graphs like contextual paragraphs. Among other works, Chen et al. (2024b) leverage LLMs both as a generator for explanations and a classifier for graph problems. Zhao et al. (2023a) propose a framework for encoding graphs into natural language, and Ye et al. (2024) extend their work by instruction tuning LLMs, but they primarily focus on node classification tasks. Müller et al. (2023); Fatemi et al. (2023); Perozzi et al. (2024) all introduce novel schemes for encoding graphs in prompts. By-and-large, these studies have tackled tasks on undirected, cyclic, static graphs, with applications geared toward standard GNN benchmarks (Hu et al., 2020a). In a parallel line of work, language models have been used to improve GNN performance. We expand on this related work in Appendix B. Extending this area of research, we focus on fine-tuning LLMs to solve learning tasks on *conversation trees*, which are directed, acyclic, and temporal graphs capable of representing a wide variety of complex human interaction sequences.

**Conversation Trees.** The ubiquity of social media has created a global shift in information consumption and human discussion (Akram & Kumar, 2017). Information on social media is frequently presented as a central post and its subsequent comments, creating a dynamic exchange between the original poster and other users. We call the graph structures projected by these posts with their corresponding comments as "conversation trees", a sub-class of text-attributed graphs (Yan et al., 2023). The study of conversation trees has been motivated by several tasks which are central themes in social media data. Some canonical problems in this domain are information flow (Bakshy et al., 2012), controversial post detection (Benslimane et al., 2021; Garimella et al., 2018), and fake news detection (Lillie & Middelboe, 2019; Han et al., 2020), among others. Similarly, bias detection (Chen et al., 2022; Zhu et al., 2022), fraud detection (Liu et al., 2023b; Zeng & Tang, 2021), event detection (Gao et al., 2021; Ji et al., 2021) and malicious behaviour detection (Wu et al., 2022; Dou, 2022) are other active research areas. Early detection (Zhou et al., 2019; Tian et al., 2020) in such cases is also critical, since it enables proactive interventions by social media platforms. Here, we propose methodology for enabling LLM competency across this entire space of problems.

## 3 METHODOLOGY

We now describe our core contribution: TREETOP, an instruction-tuning framework for LLM generalization on conversation tree learning problems. Figure 1 illustrates the high-level pipeline (see Appendix Figure 6 for a flow diagram). TREETOP consists of a novel set of structural tasks on conversation trees to fine-tune an LLM. The purpose of these tasks is to improve the LLM's ability to understand structure of conversation trees, allowing the model to follow both the content and the flow of the discussion between the constituent users.

### 3.1 REPRESENTATION OF SOCIAL MEDIA AS CONVERSATION TREES

Figure 2a shows a typical social media conversation which we directly encode as a prompt in our framework. Here, $c_0$ represents the top-level post, i.e. the root node of the conversation tree. Other posts $c_i$ are replies to either the top-level post or other replies. In our framework, we encode only the tree structure (Figure 2a) in our prompt, leaving the interaction graph (Figure 2b) to be learned through fine-tuning on structural tasks.

**Encoding.** We encode a tree node, and the entire tree from Figure 2a as follows:

$$\text{encoding(comment)} := \langle \texttt{Comment-ID, User-ID, Content, Parent, Node Features} \rangle$$

$$\text{encoding(conversation tree)} := \left[ \langle \text{encoding}(\texttt{comment}_0) \rangle \langle \text{encoding}(\texttt{comment}_1) \rangle \ldots \right]$$

For example, if we choose `timestamp` as a node feature, the tuple representation of the node corresponding to $c_2$ in Figure 2a is $\langle c_2, u_2, \texttt{-content-}, c_0, t_2 \rangle$. Multiple types of node-level features, such as "verified status" or "karma", may be available on different platforms and can also be captured in our encoding. We sort the comments by `timestamp` in the tree encoding, i.e. the main post is the first element (`comment`$_0$) in the tree encoding. We use `timestamp` to sort to mimic the natural user experience on social media platforms – a user who views a post at time $t$ can see all the comments prior to time $t$ on that post. In case a dataset doesn't provide us with the `timestamp`, we implement a breadth-first sort order in the encoding.

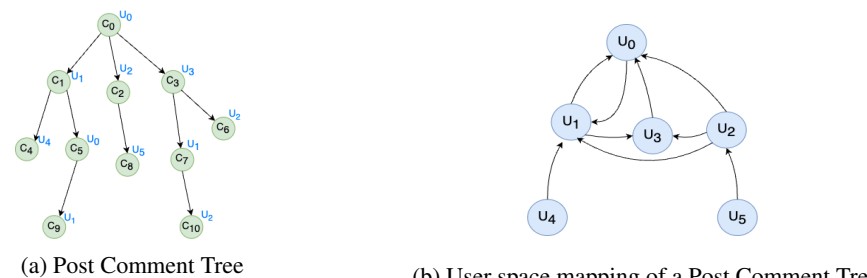

(a) Post Comment Tree        (b) User space mapping of a Post Comment Tree

Figure 2: Figure 2a represents the actual conversation tree, whereas Figure 2b shows a user-user interaction network. In TREETOP, we encode the graph in Figure 2a directly within the prompt, and TREETOP reasons over the user-user graph on its own.

## 3.2 STRUCTURAL TASKS

The core idea of TREETOP is to enhance LLM understanding of tree topology, and use this enhanced understanding alongside their inherent language understanding for downstream applications. Thus, to achieve this goal, we train LLMs on primitive topological tasks on conversation trees. Our approach is analogous to the use of graph motifs (Paranjape et al., 2017) for complex graph-based computations – we believe that LLMs will be able to compose multiple topological primitives together to solve general tree inference problems.

The primary workhorse of our framework is thus a collection of 17 reasoning tasks over conversation trees, illustrated in Figure 3. 12 of these tasks are used for fine-tuning, and the remaining used for evaluation. We designed these tasks to enable/evaluate four different "proficiency" categories on trees: (i) comment $\times$ comment tasks, that focus on the relationships between comments; (ii) user $\times$ user tasks, that focus on the relationships between users - which TREETOP-tuned LLMs infer from the conversation trees (refer Figure 2b); (iii) node characteristics tasks, that focus on the topological properties of nodes; and (iv) tree characteristics tasks, that focus on the topological properties of the entire conversation tree.

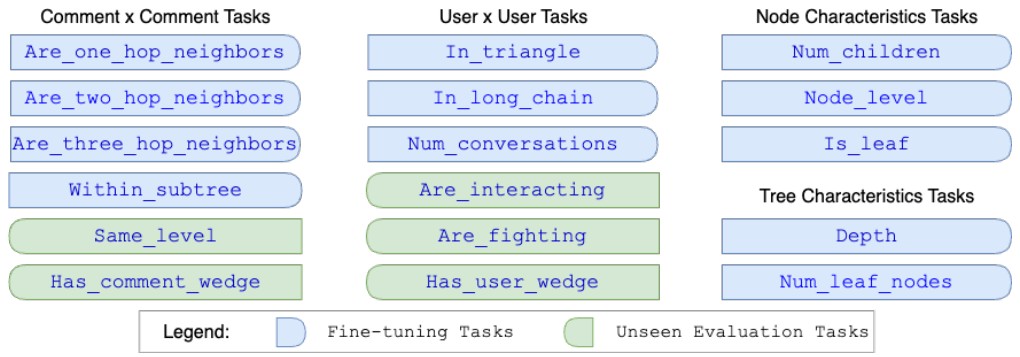

Figure 3: Structural tasks used in the TREETOP framework. These structural tasks are divided into four "proficiency" classes, as described in Section 3.2. The tasks used during fine-tuning are shown in blue, and the tasks used for evaluation are shown in green.

For instance, `Are_one_hop_neighbors` tests if one provided comment is a direct response to another provided comment. Similarly, `In_triangle` if two provided users participate in a triangular discussion with a third user, as explained in Example 1 of Section 3.2. We provide descriptions of all these tasks in Tables 7 and 8 in Appendix C.1. Each task is encoded as a `Yes` or `No` question. We design multiple prompt styles to phrase the question for each structural task. Some of these prompts use graph-topology based language (e.g. "`Whether ⟨C2⟩ has more than 3 children?`") and some use language relevant to social media platforms (e.g. "`Whether comment ⟨C2⟩ has more than 3 replies?`"). All the variations in the prompts are described in Appendix C.3.

### 3.2.1 Structural Task Corpus Creation

We created our structural task dataset using a small subset of the Pushshift Reddit data released by the authors of the Pushshift platform (Baumgartner et al., 2020), available for download here. This dataset contains all the posts and comments of Reddit that were posted in the month of April 2019. Our fine-tuning corpi are built from a random sample $S$ of $100K$ conversation trees from this dataset. For each task, we sample an equal number of positive and negative examples by mining the trees from $S$, described further in the next section. Using this approach, we are able to generate any number of labeled questions, for any task. As described further in Section 4, we fine-tune both closed-source and open-source LLMs using the TREETOP framework. We create a corpus with 10k questions per task for these experiments; we provide multiple data ablation studies on this corpus in Appendix I.

### 3.2.2 Structural Task Examples

We describe three tasks here, and detail the rest in the Appendix C.1. To sample questions for a given task, we go through our conversation tree corpus $S$, and detect the structure of interest (e.g. wedge / triangle / long chain / etc.) in each conversation tree. Each time a structure is detected, we create a positive sample, and we create an analogous negative sample (from the same tree) from node tuples that do not participate in that structure.

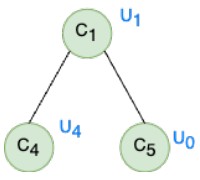 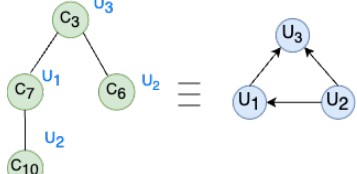 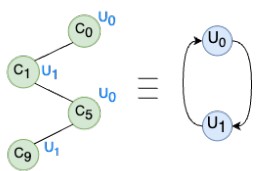

(a) User "wedge" discussion.  (b) Triangular discussion.  (c) 2-user discussion chain.

Figure 4: Illustrations for structural tasks. These figures are subtrees of the conversation tree in fig 2a. The prompts to the model only capture the conversations depicted in `green`.

**Example Task 1 - User wedge detection:**  A "wedge" is an important topological construct (Albert & Barabási, 2002), and is formed when two users respond to a single comment (see Figure 4a for illustration).

**Example Task 2 - Triangle Detection between** 3 **users:**  This task builds upon wedge detection. We define a discussion between three users A, B and C as triangular if there exists an instance where, for example, both User B and User C comment on a single comment by User A. In addition to this, if User C also comments on User B's comment (or vice-versa), a triangle is formed. See Figure 4b for an illustration.

**Example Task 3 - Existence of long chain between two users:**  We define a long chain of to and from discussion between two users A and B when there exists an instance where User B comments on a comment by User A followed by User A commenting on User B's comment to his comment and so on. See Figure 4c for illustration.

### 3.3 Structural Task Performance of Native LLMs

We evaluated different LLM model families with different model sizes on our unseen structural tasks using 100 randomly sampled instances. We report these results in Table 1, which shows that none of the existing models can successfully understand and reason over topology of these conversation trees.

---

[2]We report zero-shot results in this table because multiple of these models only afford a short context length. All models are IT variants, and are hyperlinked to their corresponding repositories.

Table 1: Results of structural tasks for zero-shot inference using different model families and model sizes.[2] We report accuracy numbers in this table. The results show that all these model families and model sizes exhibit headroom to improve understanding and reasoning over structure.

| Task | GEMMA 2B | PHI MINI | MISTRAL 7B | GEMMA 9B | GEMINI PRO | PaLM BISON |
|---|---|---|---|---|---|---|
| Same_level | 13.4 | 34.0 | 39.0 | 52.0 | 52.3 | 53.0 |
| Has_comment_wedge | 27.5 | 40.0 | 40.0 | 59.0 | 49.2 | 54.9 |
| Are_interacting | 22.1 | 52.0 | 49.0 | 65.0 | 52.1 | 72.9 |
| Are_fighting | 20.9 | 42.0 | 56.0 | 55.0 | 57.9 | 59.3 |
| Has_user_wedge | 34.7 | 40.0 | 43.0 | 53.0 | 50.8 | 54.5 |

## 4 EXPERIMENTS

We now describe experiments testing our primary hypothesis: that fine-tuning an LLM on primitive structural tasks over conversation trees enables generalization and profitable further fine-tuning on downstream tree problems. For brevity, we refer to LLMs fine-tuned with TREETOP as LLM-TTs, and others as "native" LLMs. We show that (i) LLM-TTs can decisively solve seen structural tasks and generalize to unseen structural tasks, whereas native LLMs cannot; (ii) LLM-TTs further fine-tuned on downstream, real-world tasks outperform both GNN baselines and native LLMs fine-tuned on those same tasks; and (iii) representations learned by LLM-TTs are robust and explainable.

**Fine-tuning:** Throughout this section, we analyze the impact of TREETOP fine-tuning with GEMMA-2B, PaLM-Bison, and GEMINI-PRO models. Each LLM-TT model was fine-tuned on 10k randomly-sampled tasks from each of the 12 "fine-tuning tasks" shown in 3, holding out the other 5 "unseen" tasks for out-of-distribution evaluation. We adopt the unified encoding presented in 3.1, with hardware and hyperparameter details listed in Appendix F.

### 4.1 RESULTS ON PRIMITIVE STRUCTURAL TASKS

As we showed in Table 1, native LLMs from a variety of architectures do not perform well on TREE-TOP primitive tasks. In total, we find that TREETOP fine-tuning significantly raises performance on the 12 fine-tuning tasks and even allows for generalization to the 5 tasks unseen during fine-tuning.

Specifically, Table 2 shows the complete experimental results comparing GEMINI-TT with native GEMINI on the TREETOP tasks. Experimental results with GEMMA-2B and PaLM-Bison are similar and shown in Appendix G. Overall, the collection of these results validates our hypothesis that TREETOP fine-tuning causes profitable generalization to tree tasks. First, we note that TREETOP allows GEMINI-TT to achieve **near-perfect** performance on the seen fine-tuning tasks, which shows that GEMINI-TT has indeed learned explicit reasoning skills for conversation trees. Surprisingly, GEMINI-TT also improves over GEMINI on the *unseen* TREETOP tasks, showing that these skills generalize to novel tasks. This result is strong proof-of-concept for TREETOP, and explains our results in the next section, showing that LLM-TT models generalize strongly and outperform all competitors (including both native LLMs and non-LLM state-of-the-art) on a wide variety of downstream tasks on conversation trees.

### 4.2 RESULTS ON DOWNSTREAM SOCIAL MEDIA TASKS

We chose the following collection of discussion tree classification tasks to test the application potential of LLM-TTs: (i) **Controversial post detection** (Hessel & Lee, 2019), or identification of "posts that split the preferences of a community, receiving both significant positive and significant negative feedback." (ii) **Rumor detection** using the PHEME9 dataset (Kochkina et al., 2018), (iii) **Fake news detection** using the Fakeddit dataset (Nakamura et al., 2019), and (iv) **Winning argument thread detection** (Tan et al., 2016), i.e. identification if a viewpoint of the original post author has been changed by any of the replies, collected from the r/changemyview subreddit. We provide statistics and download links for all these datasets in Appendix D.1 (Table 13) and license descriptions in Appendix L. We show that LLM-TTs fine-tuned on these tasks outperform both na-

Table 2: Results of structural tasks for zero-shot inference using GEMINI, two-shot inference using GEMINI and GEMINI-TT. GEMINI-TT is the Gemini model fine-tuned using the TREETOP framework. We show results with two-shot to provide one instance of both positive and negative class, and we couldn't test with more examples because of context length limitations. Gemini model used here is the GEMINI PRO version. Acc, Rec, Pre refer to Accuracy, Recall and Precision respectively. We highlight the best accuracy numbers for each task. Standard error for all results is reported in Appendix I.2 (Table 24).

| Fine-tuning Tasks | GEMINI (Zero-shot) | | | | GEMINI (Two-shot) | | | | GEMINI-TT | | | |
|---|---|---|---|---|---|---|---|---|---|---|---|---|
| | Acc | Rec | Pre | F1 | Acc | Rec | Pre | F1 | Acc | Rec | Pre | F1 |
| Are_one_hop_neighbors | 39.0 | 27.3 | 35.5 | 30.9 | 47.9 | 84.5 | 48.8 | 61.9 | **100.0** | 100.0 | 100.0 | 100.0 |
| Are_two_hop_neighbors | 58.1 | 48.4 | 61.0 | 54.0 | 56.2 | 93.8 | 53.6 | 68.2 | **99.9** | 99.8 | 100.0 | 99.9 |
| Are_three_hop_neighbors | 45.5 | 27.5 | 42.2 | 33.3 | 40.3 | 58.2 | 42.9 | 49.4 | **100.0** | 100.0 | 100.0 | 100.0 |
| Within_subtree | 83.3 | 84.4 | 82.9 | 83.6 | 86.3 | 90.5 | 83.5 | 86.9 | **100.0** | 100.0 | 100.0 | 100.0 |
| In_triangle | 57.6 | 38.1 | 57.2 | 45.7 | 60.4 | 63.3 | 59.8 | 61.5 | **91.6** | 92.1 | 91.1 | 91.6 |
| In_long_chain | 59.1 | 3.0 | 74.6 | 5.8 | 58.1 | 43.3 | 61.6 | 50.9 | **96.5** | 98.7 | 94.6 | 96.6 |
| Num_conversations | 60.3 | 65.6 | 59.7 | 62.5 | 64.3 | 98.3 | 58.5 | 73.3 | **99.9** | 99.9 | 100.0 | 99.9 |
| Num_children | 53.5 | 32.3 | 53.8 | 40.3 | 55.3 | 97.6 | 52.9 | 68.6 | **99.8** | 100.0 | 99.6 | 99.8 |
| Node_level | 58.1 | 77.8 | 55.5 | 64.8 | 74.8 | 87.5 | 69.8 | 77.6 | **94.5** | 99.2 | 90.7 | 94.8 |
| Is_leaf | 57.4 | 37.7 | 56.5 | 45.2 | 68.7 | 80.9 | 65.1 | 72.1 | **99.9** | 99.9 | 100.0 | 99.9 |
| Depth | 55.1 | 16.5 | 73.5 | 26.9 | 61.5 | 93.9 | 57 | 70.9 | **93.7** | 90.7 | 96.6 | 93.5 |
| Num_leaf_nodes | 56.2 | 50.5 | 56.9 | 53.5 | 51.0 | 98.8 | 50.5 | 66.8 | **87.7** | 96.7 | 81.9 | 88.7 |
| **Unseen Tasks** | | | | | | | | | | | | |
| Same_level | 52.3 | 6.2 | 76.8 | 11.4 | 47.5 | 28.7 | 46 | 35.3 | **76.1** | 81.1 | 73.6 | 77.1 |
| Has_comment_wedge | 49.2 | 10.6 | 47.3 | 17.3 | 53.5 | 83.3 | 52.2 | 64.2 | **63.0** | 78.8 | 59.8 | 68.0 |
| Are_interacting | 52.1 | 6.2 | 78.1 | 11.5 | 70.0 | 73.6 | 68.7 | 71.0 | **78.2** | 59.1 | 95.7 | 73.0 |
| Are_fighting | 57.9 | 16.1 | 84.2 | 27.0 | 68.4 | 51.5 | 77.8 | 62.0 | **86.2** | 99.9 | 78.4 | 87.9 |
| Has_user_wedge | 50.8 | 12.7 | 50.9 | 20.3 | 51.3 | 54.8 | 51.2 | 53.0 | **61.6** | 43.3 | 68.3 | 53.0 |

tive LLMs (likewise fine-tuned) and non-LLM state-of-the-art, showing the promise of TREETOP toward important applications. We discuss results from tasks (i) and (iv), leaving discussions of (ii) and (iii) to Appendix E.

### 4.2.1 EXPERIMENTAL DESIGN

We divided each dataset into a random 70:15:15 split for fine-tuning, validation, and testing. The validation set was used to select the best LLM checkpoint from the fine-tuning phase. We use 5-way cross-validation with a bootstrapping approach to derive test-set standard errors, described fully in Appendix I.2.2. During fine-tuning, each LLM receives each discussion tree input in the TREETOP encoding (see Section 3.1), along with a yes/no prompt: `"Whether the post is a [X] post?"`, where `[X]` is `controversial`, `rumor`, `fake news`, or `winning argument`, depending on the data set. Across our different experiments, we compare five models: GEMINI (zero-shot and two-shot), GEMINI fine-tuned for that specific social media task, GEMINI-TT (zero-shot), and GEMINI-TT fine-tuned for that specific social media task. We also compared with 3 GNN-baselines of GCN, GAT and GraphSage where applicable. For the GNN baselines, we use the BERT model to embed the text content of posts and comments in the conversation tree. Additionally, we also provide the state-of-the-art GNN-based benchmark for comparison. We report the results for Controversial post detection and Winning argument thread detection tasks in the sections below and results for Rumor and Fake News detection are reported in Appendix E. Additionally, we perform another ablation study comparing two different encodings for the tree on the downstream tasks, as reported in Appendix H.

### 4.2.2 CONTROVERSIAL POST DETECTION RESULTS

Table 3 presents controversial post detection results across all models, including the DFE-GCN algorithm (Hua et al., 2023), the prior state-of-the-art GNN-based model that combines sentence-BERT

with GCNs.[3] GEMINI-TT fine-tuned for this task has the highest performance across all metrics, even exceeding GNN baselines and DFE-GCN. We note that this dominant performance is due both to topology-aware capacities gained from TREETOP, as well as fine-tuning on this particular task. TREETOP's effect can be seen by the marked improvement of GEMINI-TT 0-shot F1 against GEMINI 0-shot F1. Fine-tuning GEMINI-TT on this task is then much more efficient, leading to state-of-the-art.

Table 3: Results for controversial post detection task across all models. "ZS" means zero-shot, "2S" means two-shot, and "FT" refers to fine-tuning on the detection task. GSG refers to the GraphSage GNN baseline. Standard error for all results is reported in Appendix I.2 (Table 25).

| | GEMINI | | | GEMINI-TT | | GNN Baselines | | | SOTA |
|---|---|---|---|---|---|---|---|---|---|
| **Metric** | **ZS** | **2S** | **FT** | **ZS** | **FT** | **GCN** | **GAT** | **GSG** | **DFE-GCN**[3] |
| Acc | 50.0 | 56.4 | 68.6 | 50.6 | **84.6** | 64.0 | 68.0 | 66.0 | 76.6 |
| Rec | 27.4 | 53.4 | 85.9 | 90.5 | 87.7 | 92.0 | 75.0 | 85.0 | 67.2 |
| Pre | 58.2 | 56.9 | 64.0 | 50.4 | 82.6 | 59.0 | 66.0 | 62.0 | 67.4 |
| F1 | 37.3 | 55.1 | 73.3 | 64.7 | 85.0 | 72.0 | 70.0 | 72.0 | 67.3 |

### 4.2.3 WINNING ARGUMENT THREAD DETECTION RESULTS

In Table 4 we give the results for the winning argument thread detection task. We compare with the approach in Tan et al. (2016), combining multiple linguistic and interaction-based features of conversation trees, comprising the most recent prior benchmark.[3] GEMINI-TT outperforms Tan et al. and GEMINI across all metrics. Even zero-shot performance of GEMINI-TT is better than GEMINI *fine-tuned* for this specific task. We attribute this result to the enhanced topological understanding brought about by TREETOP fine-tuning.

Table 4: Results for the winning argument thread detection task. Standard error for all results is reported in Appendix I.2 (Table 26).

| | GEMINI | | | GEMINI-TT | | SOTA |
|---|---|---|---|---|---|---|
| **Metric** | **ZS** | **2S** | **FT** | **ZS** | **FT** | **Tan et al. (2016)**[3] |
| Acc | 50.2 | 47.3 | 51.5 | 52.5 | **76.6** | 70.0 |
| Rec | 3.2 | 5.6 | 12.3 | 45.2 | 83.5 | - |
| Pre | 51.3 | 60.0 | 57.5 | 53.0 | 73.4 | - |
| F1 | 6.1 | 10.3 | 20.3 | 48.8 | 78.1 | - |

### 4.2.4 EARLY DETECTION RESULTS

Early detection of future conversation properties is a significantly important challenge with real-world impact (Akram & Kumar, 2017), given the ubiquity of social media platforms. In Figure 5, we report results of zero-shot early detection of controversial posts for both GEMINI and GEMINI-TT *fine-tuned* for the controversial posts detection task. For each dataset, we take views of each conversation tree at different timestamps, where a "view" at timestamp $t$ contains the original post and all replies up to time $t$. We use $t = 0, 1, 2, 4, 6, 12, 24$ and `inf` hours in our experiments. Appendix D.2 shows the statistics on what fraction of comments are seen at different time durations.

The fine-tuned GEMINI-TT achieves the same performance of fine-tuned GEMINI 20 hrs in advance. Specifically, GEMINI's accuracy at 24 hrs is 68.6% - GEMINI-TT has an accuracy of 62.6% and 68.9% at 2hrs and 4hrs respectively. Additionally, GEMINI-TT achieves 73.7% accuracy at 6hrs, and 83.3% accuracy at 24hrs. These results show that these models can be effective at early detection of controversial post in just 4 to 6 hours.

---

[3]These are results as reported in Hua et al. (2023) and Tan et al. (2016), acknowledging that the inputs and test sets might not have parity. (Tan et al., 2016) only provide accuracy, and precision/recall/f1 are not reported in the paper.

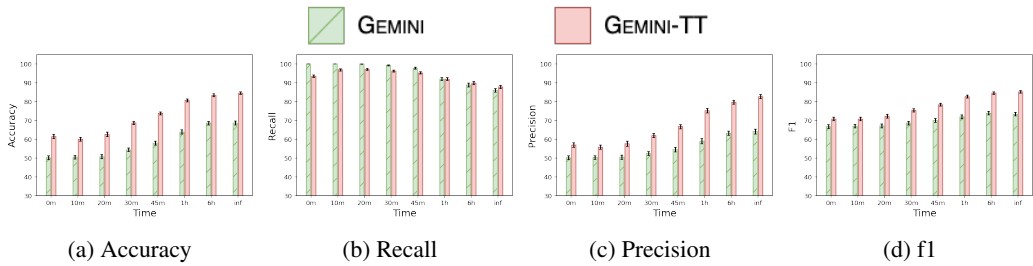

(a) Accuracy          (b) Recall          (c) Precision          (d) f1

Figure 5: These graphs show the early-detection results comparing the versions of GEMINI and GEMINI-TT both *fine-tuned* for detection of controversial post detection tasks. Error bars are shown in the plot and all the standard errors for model accuracy are less than 1.3.

We provide the same results on rumor detection in Appendix E.3, we see similar dominance of fine-tuned GEMINI-TT over fine-tuned GEMINI there as well.[4]

### 4.3 EXPLANABILITY AND ROBUSTNESS OF REPRESENTATIONS LEARNED BY LLM-TTs

In this section we empirically examine the post-fine-tuning regression, learnt embeddings, and ablations of TREETOP.

#### 4.3.1 REGRESSION ANALYSIS

We ran MMLU and GSM8K benchmarks comparing both zero-shot GEMINI and zero-shot GEMINI-TT to understand the regression loss incurred by the native LLM due to TREETOP. We show the model accuracies in Table 5. The results show that TREETOP fine-tuning does not incur any substantial loss in LLM performance as measured against the MMLU and GSM8K benchmarks. This serves as an empirical stamp of robustness for the TREETOP framework.

Table 5: Evaluation of regression loss (model accuracy is reported).

| Benchmark | GEMINI | GEMINI-TT |
|-----------|--------|-----------|
| MMLU      | **65.65** | 65.47 |
| GSM8K     | **83.5**  | 83.0  |

#### 4.3.2 ANALYSIS OF TREETOP EMBEDDINGS

To examine the explainability of models tuned with the TREETOP framework and provide additional evidence that it has indeed learned tree topologies, we computed embeddings from PaLM-TT from a subset of our data.[5] In particular, by computing embeddings both with and without the text content in the conversation trees, we were able to isolate the embedding of the topology of each tree. We found that these topology-focused embeddings exhibited strong topology-aligned clustering and that a classifier trained purely on topological embeddings obtains a 77% accuracy on a topology classification task using a logistic regression classifier. We describe these results in full in Appendix J.

#### 4.3.3 ABLATIONS AND ROBUSTNESS

To further demonstrate the robustness and repeatability of our work, we performed a variety of ablation studies and similar exercises. We describe these all in full in Appendix G and Appendix I. Briefly, we have shown the following: (i) GEMINI-TT benefits slightly across-the-board from more structural task fine-tuning examples (we tried 100k vs the 10k used throughout this section); (ii) test-set bootstrapping reveals low standard errors of all metrics presented throughout this section, demonstrating high repeatability of our experiments; (iii) repeating TREETOP fine-tuning with four different disjoint training corpora revealed that the results in this section are extremely stable; (iv) 5-fold cross-validation over all four downstream social media tasks reveal robust metric stability.

---

[4]We do not evaluate early detection on the other two tasks, as the Fakeddit dataset does not provide timestamps, and the winning argument thread detection task requires the entire thread.

[5]Embeddings from GEMINI models are not accessible.

Furthermore, while the results in this section focus on GEMINI, the results of GEMMA and PaLM-Bison are provided in Appendix G.

## 5  DISCUSSION

In this paper, we introduced TREETOP, a framework and suite of tasks for fine-tuning LLMs to perform well on topological reasoning tasks over conversation tree inputs. As we showed through our experiments, LLMs fine-tuned with TREETOP generalize significantly better to out-of-distribution reasoning tasks, and have achieved state-of-the-art on a variety of downstream tasks in various social media domain applications. We now address limitations and future directions.

A limitation of our study is that the vast majority of our datasets, including the entirety of our primitive structural tasks, are derived from Reddit data. Only one of our downstream task datasets comes from a different platform (namely X). While this limits our paper's empirical insights, we believe that our strong generalization results shows the promise of the TREETOP framework to allow profitable fine-tuning of LLMs across other social platforms and even other platform types such as Q&A forums and collaborative editing (Daxenberger & Gurevych, 2012). We also note that models such as those fine-tuned on TREETOP are often used for content-moderation (Roberts, 2017; Gillespie, 2020). A serious risk for such models centers around their potential misuse for content over-moderation, thereby promoting echo-chambers and insularity of thought (Kumar et al., 2024). Practitioners and platform designers should ensure that auto-moderation models trained with TREETOP (or any similar framework) are deployed responsibly, such as preferring "soft" moderation tags (shown in Martel & Rand (2023) to be effective) over bans/takedowns, providing definitional clarity to moderation policies, and offering transparency into moderation decisions, such as open-sourcing logs (Macdonald & Vaughan, 2024; Singhal et al., 2023). Additionally, our experiments show that TREETOP models are performant at early-detection (68.9% accuracy at 4 hours), which can enable effective human-in-the-loop moderation systems to be designed (Lai et al., 2022).

As LLMs become better-able to handle multimodality (Zhang et al., 2024), an important next step in this line of research is to allow large models to learn signals from the joint distribution of topology and non-text modes such as images and videos. Future efforts in this direction should be inspired and motivated by existing non-LLM work in multimodal cascade prediction (Xie et al., 2020; Zhang et al., 2018), and by the importance of multimedia content in such applications (Nakamura et al., 2019). Aside from multi-modality, another interesting future direction would be designing primitive tasks defined over *multiple* trees, involving resolving same-user behavior and cross-dependencies, as a way to fine-tune models for more complex longitudinal effects. More generally, we hope that the concepts underlying TREETOP– fine-tuning on structural tasks after pre-training on language tasks – may be a re-usable recipe for teaching LLMs to become performant on structured data.

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

# A  SUPPLEMENTARY: TABLE OF CONTENTS

We first present the Table of Contents for the supplementary material in this Appendix.

Table 6: List of supplementary content in this Appendix.

## B Supplementary: Related Work

One of the most seminal advances that has spurred the current wave of LLM research has been the attention mechanism, and the transformer architecture (Vaswani et al., 2017). This led to development of newer language encodings, notably BERT (Devlin et al., 2018). Radford et al. (2018) popularized generative pre-training followed by discriminative fine-tuning for downstream tasks - an approach that is highly leveraged in the application of LLMs. Raffel et al. (2020) introduced the T5 paradigm, a unified framework to convert all text-based language problems into a text-to-text format. This work also inspired modeling graph problems into text prompts. Wei et al. (2021) provided an instruction-tuning framework to fine-tune LLMs for improved for zero-shot performance, whereas Brown et al. (2020) showed an improvement in LLM performance under few-shot training. Given this success of LLMs on language understanding, they have been subsequently been applied to other modalities as well, notably images (Alayrac et al., 2022), and now to the creation of multimodal models (Team et al., 2023; Huang et al., 2024; Reid et al., 2024).

Similarly, graph neural networks (Scarselli et al., 2008) were introduced for machine learning on graphs, and their expressive power has been extensively studied theoretically (Xu et al., 2018) and across multiple benchmarks (Chien et al., 2021). Extensive surveys (Wu et al., 2020; Zhou et al., 2020; Chami et al., 2022) on GNNs demonstrate that they have been a prominent approach to reason over graphs. Graphs have been successfully applied to a variety of problem domains, for instance, cryptography (Priyadarsini, 2015), transportation networks (Bisen, 2017), quantum physics (Gilmer et al., 2020), chemistry (Gilmer et al., 2017; Hu et al., 2020b), and in multiple domains of everyday use and technology (Kaundal, 2017). For example, knowledge graphs have seen widespread use and adoption in retrieval and query-based systems (Schlichtkrull et al., 2018; Yasunaga et al., 2022a).

Multiple recent studies have leveraged cross-pollination of ideas between GNNs and LLMs. Zhang et al. (2023b) look at solving spatio-temporal problems on dynamic graphs using large language models. Zhang (2023) notes that using tools can help an LLM reason over graph structures. On the other hand, Yasunaga et al. (2022b) view a text corpus as a graph of documents, and propose LinkBERT, an LM pre-training method that leverages links between documents. Similarly, Zhu et al. (2021) and Qin et al. (2023) propose methods that use GNNs to improve the performance of a text encoder. Alternatively, Duan et al. (2023) use LLM embeddings to initialize the GNN node features, and along with Perozzi et al. (2024), contribute to the area of work which integrates LLMs and GNNs at the level of tokens and embeddings. In a parallel line of work, language models have been used to improve GNN performance. Xu et al. (2023) use a BERT model to encode textual features on nodes whereas He et al. (2023) leverage explanations from an LLM to improve performance of downstream GNNs.

Similarly, in the domain of social media analysis, multiple studies have been performed that discuss the impacts of consuming social media information on the individual and the population (Amedie, 2015; Bail et al., 2018; Akram & Kumar, 2017; Olan et al., 2024). Conversation trees are central to how information is consumed in social media – there is a central post that an original poster makes, and there are comments in response to this post. Information flow on social media has motivated several lines of work (Lerman & Ghosh, 2010; Bakshy et al., 2012; Liu et al., 2023a; Pröllochs & Feuerriegel, 2023; Hardy et al., 2023). Controversial post detection is of central importance to ensure that social media users are not exposed to potentially harmful content (Benslimane et al., 2021; Madhu et al., 2023; Garimella et al., 2018; Qiu et al., 2019). Similarly, a growing body of research is dedicated to addressing the issue of fake news, with studies exploring various detection methods and their effectiveness (Han et al., 2020; Shu et al., 2017; Nakamura et al., 2019; Lillie & Middelboe, 2019; Ma & Gao, 2020; Dou et al., 2021). Multilingual generalization (Li & Li, 2022; Wen, 2023; Li et al., 2022), bias detection (Olteanu et al., 2019; Chen et al., 2022; Zhu et al., 2022), fraud detection (Liu et al., 2023b; Zeng & Tang, 2021; Chen et al., 2024a), event detection (Abagissa et al., 2024; Gao et al., 2021; Ji et al., 2021) and malicious behaviour detection (Wu et al., 2022; Toshevska et al., 2023; Dou, 2022) are other important problems in this area.

## C  SUPPLEMENTARY: METHODOLOGY

Figure 6 represents the pipeline for the creation of models using the TREETOP framework. We first begin here by providing the detailed listing of all the structural tasks, and follow that up with the prompts used for each task.

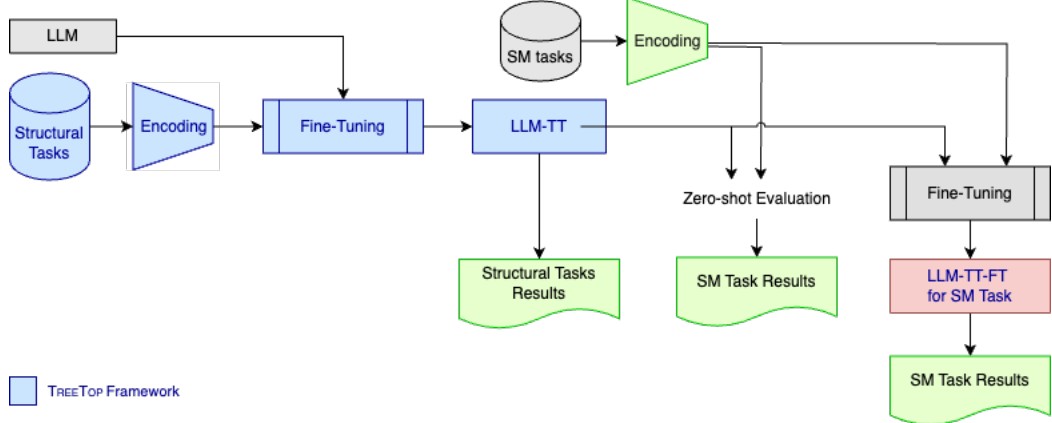

Figure 6: TREETOP's Process Flow Diagram. The blue boxes represent the core TREETOP framework. SM stands for Social Media, and is generalizable to all conversation tree tasks.

### C.1  STRUCTURAL TASKS DESCRIPTION

We provide the descriptions of all the 17 tasks provided in the TREETOP framework in Table 7 and Table 8. The four classes of tasks are described in Section 3.2.

Table 7: Structural tasks and their explanation. The set of tasks are clustered into 4 categories with different semantic application to conversation trees. This table shows the tasks TREETOP is trained on.

| **Tasks for fine-tuning** | |
|---|---|
| **Comment $\times$ comment tasks** | |
| Are_one_hop_neighbors | Are two comments one-hop neighbors of each other |
| Are_two_hop_neighbors | Are two comments two-hop neighbors of each other |
| Are_three_hop_neighbors | Are two comments three-hop neighbors of each other |
| Within_subtree | Is a comment in the subtree rooted at another given comment |
| **User $\times$ user tasks** | |
| In_triangle | Does there exist a triangular discussion between three users |
| In_long_chain | Does there exist a to-and-fro conversation between two users |
| Num_conversations | Given a user, are more than $k$ comments by the user in the tree |
| **Node characteristics tasks** | |
| Num_children | Given a comment, are more than k replies to it |
| Node_level | Given a node, detect its level |
| Is_leaf | Is a given node a leaf node i.e. it has zero replies |
| **Tree characteristics tasks** | |
| Depth | Given a post-comment tree, detect its depth |
| Num_leaf_nodes | Are there more than $k$ nodes in the tree with zero replies |

Table 8: This table is similar to Table 7, except it lists the set of unseen tasks that are used to assess TREETOP's structural understanding.

| **Unseen tasks for evaluation** | |
| --- | --- |
| **Comment $\times$ comment tasks** | |
| Same_level | Given a comment pair, whether they are at same level |
| Has_comment_wedge | Whether the given user pair reply to a same comment |
| **User $\times$ user tasks** | |
| Are_interacting | Given a user pair $(U_1, U_2)$, whether $U_1$ replies to $U_2$'s comment |
| Are_fighting | Given a user pair $(U_1, U_2)$, whether $U_1$ comments on $U_2$ more than 2 times |
| Has_user_wedge | Given three users $(U_1, U_2, U_3)$, whether $U_1$ replies to both $U_2$ and $U_3$. |

## C.2 PROMPT ENCODING

The TREETOP framework presents an encoding which captures both the topology and content of conversation trees. This encoding captures the content of both the main post and comments along with the structure of the comments tree for that post. Here, each element in the encoding of the conversation tree has all the richness that we can provide from the dataset.

The complete prompt employed by the TREETOP framework is:

```
Given is a social media conversation tree, where each comment
(node) in the tree is of the following structure:  (<node_id>
<user_id> <content> <parent_id>).  The first node (<C0>) is the
main post on Reddit followed by the comments to the main post.
<parent_id> refers to the comment/post to which the current
comment is a reply to.

{Explanation}

QUESTION: {question}

OPTIONS:
- Yes
- No

POST-COMMENT TREE: ⟨encoding(comment₀)⟩⟨encoding(comment₁)⟩...
```

Here, {question} is the task posed as a Yes|No question. If some additional explanation of the question is required, it is added in the {Explanation} section. For example, we first define what a triangular discussion is in a conversation tree and then ask the question about it. Detailed prompts are provide in the next section.

## C.3 INSTRUCTION PROMPTS FOR STRUCTURAL TASKS

We now provide the specific details for the prompts used for all our tasks in four tables - Table 9 to Table 12 for all the four categories of structural tasks.

Table 9: This table specifies all the prompts used for all the comment $\times$ comment tasks.

| **Prompts for comment $\times$ comment tasks** |
|---|
| `Are_one_hop_neighbors`
1.  whether `<C{node2}>` is a direct comment to `<C{node1}>` ?
2.  whether `<C{node2}>` is a one-hop neighbor to `<C{node1}>` ? |
| `Are_two_hop_neighbors`
1.  whether `<C{node2}>` is a comment to one of the direct comment to `<C{node1}>` ?
2.  whether `<C{node2}>` is a two-hop neighbor to `<C{node1}>` ? |
| `Are_three_hop_neighbors`
1.  whether `<C{node2}>` is a three-hop neighbor to `<C{node1}>` ? |
| `Within_subtree`
1.  whether `<C{node2}>` is in the subtree rooted at `<C{node1}>` |
| `Same_level`
1.  whether `<C{node1}>` and `<C{node2}>` are at the same level ? |
| `Has_comment_wedge`
1.  whether users `<U{user1}>` and `<U{user2}>` reply to a same comment ?
2.  whether users `<U{user1}>` and `<U{user2}>` reply to `<C{node}>` ? |

Table 10: This table specifies all the prompts used for all the user × user interaction tasks.

| Prompts for user × user tasks |
|---|
| In_triangle
1. whether users <U{user1}> <U{user2}> and <U{user3}> are involved in a triangular discussion between each other?
2. whether there is a triangular discussion between three users ?
**Explanation:** We define a discussion between three users A, B and C as triangular if there exists an instance where lets say User B comments on a comment by User A and User C also comments on the same comment by User A and one of User B or C comments on each other's comment on User A's comment. |
| In_long_chain
1. whether users <U{user1}> and <U{user2}> are involved in a long chain of to and fro discussion of atleast length {chain_length} between each other ?
2. whether there is a long chain of to and fro discussion of at least length {chain_length} between two users ?
**Explanation:** We define a long chain of to and fro discussion between two users A and B when there exists an instance where User B comments on a comment by User A followed by User A commenting on User B's comment to his comment and so on. |
| Num_conversations
1. whether there are more than {num_comments} comments by <U{user}> ?
2. whether there are multiple comments by <U{user}> ? |
| Are_interacting
1. whether users <U{user1}> and <U{user2}> interact with each other i.e. one of them replies to other's comment ? |
| Are_fighting
1. whether user <U{user1}> replies to user <U{user2}> more than two times ? |
| Has_user_wedge
1. whether there is a user that replies to both user <U{user1}> and <U{user2}> ?
2. whether user <U{user}> replies to both user <U{user1}> and <U{user2}> ? |

Table 11: This table specifies all the prompts used for all the node characteristics tasks.

| Prompts for node characteristics tasks |
|---|
| Num_children
1. whether there are more than {num_comments} direct comments to <C{node}> ?
2. whether there are more than {num_comments} children of <C{node}> ?
3. whether there are more than {num_comments} one-hop neighbors of <C{node}> ? |
| Node_level
1. whether <C{node}> is at level {level} ? (Assuming the root node is at level 0) |
| Is_leaf
1. whether <C{node}> is a leaf node ?
2. whether <C{node}> has zero children ?
3. whether <C{node}> has no replies ? |

Table 12: This table specifies all the prompts used for all the tree characteristics tasks.

| Prompts for tree characteristics tasks |
|---|

```
Depth
1.  whether the depth of the tree is {depth} ?
2.  whether the depth of the tree is more than {depth} ?
Num_leaf_nodes
1.  whether there are more than {num_leaf_nodes} leaf nodes in the
    given tree ?
2.  whether there are more than {num_leaf_nodes} nodes in the given
    tree that have zero replies ?
3.  whether there are more than {num_leaf_nodes} nodes in the given
    tree that have zero children ?
4.  whether there are {num_leaf_nodes} leaf nodes in the given tree ?
5.  whether there are {num_leaf_nodes} nodes in the given tree that
    have zero children ?
6.  whether there are {num_leaf_nodes} nodes in the given tree that
    have zero replies ?
```

## D  DATASET STATISTICS

### D.1  SOCIAL MEDIA DATASET STATISTICS

We used popular social media datasets for the evaluation of downstream tasks. The sources and statistics for the social media datasets are presented in Table 13. The winning argument thread dataset collected data from the `r/changemyview` subreddit. Note that all the datasets are not evenly balanced between positive and negative samples, and we do not employ any under-sampling or over-sampling techniques to artificially balance the dataset. Similar to the creation of TREETOP, we used a $70 : 15 : 15$ split ratio to create training, validation and test datasets for each downstream social media task. The license descriptions for all the datasets are given in Appendix L.

Table 13: Datasets and their statistics.

| Dataset (Download Links) | Paper | Source | +ves | -ves |
|---|---|---|---|---|
| Controversial Post [link] | Hessel & Lee (2019) | Reddit | 7515 | 7518 |
| PHEME9 (rumor detection) [link] | Kochkina et al. (2018) | X | 1616 | 3058 |
| Fakeddit [post] [comments] | Nakamura et al. (2019) | Reddit | 75215 | 172371 |
| Winning Argument Thread [link] | Tan et al. (2016) | Reddit | 6557 | 6557 |

### D.2  EARLY DETECTION IN RUMOR AND CONTROVERSY DATASETS

The following table shows how much of the conversation tree is available at different timestamps to facilitate our early detection task variation. This table shows what fraction of comments are observed within the first hour, first two hours, and so on (assuming inf time is 100%).

## E  SUPPLEMENTARY DOWNSTREAM TASK EXPERIMENTS

For **controversial post detection**, we use the task definition and the dataset provided in Hessel & Lee (2019). The task is to detect Reddit "posts that split the preferences of a community, receiving both significant positive and significant negative feedback." This dataset only has posts which have at least 30 comments in the conversation tree. Similarly, for **rumor detection**, we use the PHEME9 dataset from Kochkina et al. (2018) for this task. In this work, the authors have collected, identified and annotated X posts as rumors associated with newsworthy events. Additionally, for **fake news detection**, we use the Fakeddit dataset from Nakamura et al. (2019) for this task. In this work, the authors have collected and annotated Reddit posts as being fake news or not. The results of

Table 14: This table shows the percentage of comments posted within the time elapsed since the main post.

| Controversy Detection Dataset | | Rumor Detection PHEME9 Dataset | |
|---|---|---|---|
| Time Duration | Percentage of comments received | Time Duration | Percentage of comments received |
| 0 | 0.00 | 0 | 0.00 |
| 1 hr | 12.13 | 10 mins | 34.00 |
| 2 hrs | 21.29 | 20 mins | 48.18 |
| 4 hrs | 38.23 | 30 mins | 55.74 |
| 6 hrs | 51.92 | 45 mins | 61.33 |
| 12 hrs | 76.63 | 1 hr | 68.16 |
| 24 hrs | 93.18 | 6 hrs | 89.66 |
| inf | 100.00 | inf | 100.00 |

controversial post detection have already been provided in Section 4.2.2; we now provide results for rumor detection in Appendix E.2 and fake news detection in Appendix E.1.

## E.1 FAKEDDIT

Table 15 shows the results of fake news detection on the Fakeddit dataset (Nakamura et al., 2019). Fakeddit is a multi-modal Reddit dataset and contains both text and images. For our experiments, we only use those samples that have at least 5 comments in the conversation tree while performing fine-tuning. For our algorithms, we ignore any image content in the posts and comments. The statistics for this dataset are in Table 13, and show approximately a 1:2 class imbalance between positive and negative samples. As before, we compare the performance of GEMINI (zero-shot), GEMINI fine-tuned on Fakeddit, GEMINI-TT zero-shot, and GEMINI-TT further fine-tuned on Fakeddit. In addition, we also show results for the P and PC encodings for GEMINI. We also compare our approaches with Nakamura et al. (2019), who combine BERT-encodings with a ResNet-50 image encoding model. Our results show that fine-tuned GEMINI-TT (accuracy of 96.0%) outperforms both fine-tuned GEMINI (accuracy of 89.8%) and the approach by Nakamura et al. (2019) (accuracy of 89.1%).[6] The standard error for our experiments is reported in Appendix I.2.

Table 15: Results for fake news detection task. The results in bold shows that fine-tuned GEMINI-TT beats the performance of both fine-tuned GEMINI and the approach by Nakamura et al. (2019). Also, we do not have zero-shot and fine-tuned variants for Nakamura et al. (2019). Standard error for this evaluation is provided in Appendix I.2.2 (Table 27) and are below 0.2 for accuracy for all models.

| Metric | GEMINI | | | GEMINI-TT | | GNN Baselines | | | SOTA |
|---|---|---|---|---|---|---|---|---|---|
| | ZS | 2S | FT | ZS | FT | GCN | GAT | GraphSAGE | Nakamura et al. (2019)[6] |
| Acc | 76.3 | 63.6 | 89.8 | 77.2 | **96.0** | 77.0 | 79.0 | 81.0 | 89.1 |
| Rec | 3.9 | 52.9 | 82.7 | 2.8 | 88.6 | 71.0 | 78.0 | 77.0 | - |
| Pre | 34.9 | 67.3 | 75.1 | 56.8 | 93.5 | 55.0 | 57.0 | 61.0 | - |
| F1 | 7.1 | 59.3 | 93.5 | 5.3 | 91.0 | 62.0 | 66.0 | 68.0 | - |

## E.2 RUMOR DETECTION

Table 16 shows the performance on rumor detection PHEME9 dataset (Kochkina et al., 2018). The statistics for this dataset are in Table 13, and show approximately a 1:2 class imbalance between positive and negative samples. For our experiments, we only use those samples that have at least 5 comments in the conversation tree while performing fine-tuning. As before, we compare the

---

[6]These are results as reported in Nakamura et al. (2019), acknowledging that their and our test sets might not have parity. We only report accuracy since it is the only metric reported in the paper.

performance of GEMINI (zero-shot, and two-shot), GEMINI fine-tuned for rumor detection, GEMINI-TT (zero shot) and GEMINI-TT fine-tuned for rumor detection. We also compare these with GNN baselines. Additionally, we also compare our models with two prior published research: (i) PHAROS algorithm (Nguyen et al., 2024), which integrates label information with graph homophily measures, and is among the state-of-the-art published result in this domain[7]; and (ii) NRA MOS-GAT algorithm (Patel et al., 2022), which uses oversampling and BERT embeddings along with an attention-based GNN model.[7] Our results show that fine-tuned GEMINI-TT model (accuracy of $87.1\%$) outperforms all three of fine-tuned GEMINI (accuracy of $72.5\%$), PHAROS (accuracy of $75.9\%$) and NRA MOS-GAT (accuracy of $78.4\%$). The standard error for all the results is below $4\%$, except for the precision number for GEMINI-TT-FT which had an error of around $15\%$ (computation methodology is described in Appendix I.2).

Table 16: Results for rumor detection task. The results in bold show that fine-tuned GEMINI-TT beats the performance of fine-tuned GEMINI, PHAROS, and NRA MOS-GAT. Given PHAROS and NRA MOS-GAT are not LLM based, we do not have zero-shot and fine-tuned variants for them. Standard error for this evaluation is provided in Appendix I.2.2 (Table 28) and are below $1.9$ for accuracy for all models.

| | GEMINI | | | GEMINI-TT | | GNN Baselines | | | SOTA | |
| Metric | ZS | 2S | FT | ZS | FT | GCN | GAT | GraphSAGE | PHAROS[7] | NRA MOS-GAT[7] |
|---|---|---|---|---|---|---|---|---|---|---|
| Acc | 64.0 | 54.8 | 72.5 | 65.9 | **87.1** | 78.0 | 80.0 | 79.0 | 75.9 | 78.4 |
| Rec | 6.1 | 76.2 | 47.1 | 3.1 | 78.1 | 80.0 | 71.0 | 73.0 | - | - |
| Pre | 41.3 | 41.9 | 63.7 | 76.5 | 83.6 | 65.0 | 71.0 | 69.0 | - | - |
| F1 | 10.6 | 54.1 | 54.1 | 5.8 | 80.7 | 72.0 | 71.0 | 71.0 | 77.9 | 73.1 |

We acknowledge that the zero-shot performance (recall and f1) for both GEMINI and GEMINI-TT is sub-par. It is quite likely that these models considers the tree topology content of the TREETOP encoding as noise for this task. This may also be because our prompt is simply asking these models `"whether the post is a rumor?"`, only relying on their innate language understanding of the word 'rumor' independent of any other context. In contrast, the performance of the fine-tuned variants of both GEMINI and GEMINI-TT see a major boost once they are able to learn our definition of 'rumor' from the fine-tuning dataset.

### E.3 EARLY DETECTION OF RUMORS

Similar to the early detection of controversial posts described in Section 4.2.4, we collect and evaluate fine-tuned GEMINI and fine-tuned GEMINI-TT at $t = 0, 10$ min, $20$ min, $30$ min, $45$ min, $1$ hr, $6$ hr and `inf` hours after the original post respectively for the rumor detection task. Our result shows that fine-tuned GEMINI-TT (accuracy of $87.0\%$ at $t = 0$) outperforms fine-tuned GEMINI (accuracy of $73.7\%$ at $t = 24$hrs). This shows that models fine-tuned with the TREETOP framework can also be used for extremely early detection of rumors. We also see that for all models, the performance stays quite flat over the different timespans after $t = 20$ mins. We hypothesize that this is because in the PHEME9 dataset, most of the comments appear in the first hour itself (see Table 14), and that this dataset is such where the main post itself is highly discriminative of rumors.

## F MODEL HYPER-PARAMETERS AND HARDWARE DETAILS

The base LLM used in most of our experiments was GEMINI (Team et al., 2023). We use the GEMINI model available on the Google Cloud API (Cloud, 2023), with a learning rate of 5e-7 and inferred with a temperature of 0. The input token length was set to 8196 tokens, and output token length was capped at 512 tokens (our outputs were binary `Yes | No` responses) - if our input prompt exceeded 8196 tokens, in accordance with our desire to capture the complete tree, we kept the complete main post but restricted the number of words in other comments to 12 words (this

---

[7]These are results as reported in PHAROS (Nguyen et al., 2024) and NRA MOS-GAT algorithm (Patel et al., 2022), acknowledging that their and our test sets might not have parity. We only report accuracy and f1 since those are the only reported metrics in the papers; precision and recall have not been reported.

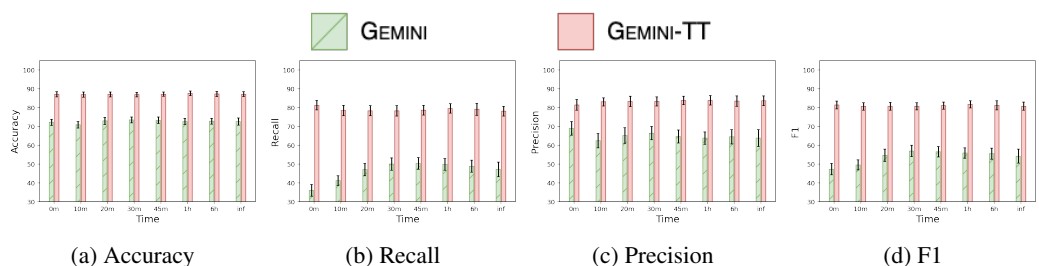

(a) Accuracy      (b) Recall      (c) Precision      (d) F1

Figure 7: These graphs show the results comparing fine-tuned GEMINI and fine-tuned GEMINI-TT for early detection of rumor detection tasks. The error bars are also shown in the column charts, and all error bars are within 1.9 (for accuracy).

truncation was not needed in any of our structural tasks, and for about 5% of the samples in the downstream tasks). Our batch size for all the experiments was set to 128. The number of steps vary across our experiments depending on the size of the dataset - GEMINI was fine-tuned for 3000 steps for the creation of GEMINI-TT. Similarly, GEMINI-TT was further fine-tuned for 500 steps for every downstream social media task other than Fakeddit, for which we fine-tuned it for 1500 steps. Each experiment was run on 512 TPU v3 chips. Every 100 steps of the experiment took approximately 1 hour to complete.

# G  MODEL ABLATIONS

We also changed the base LLM from GEMINI to GEMMA and PaLM-Bison to demonstrate that the TREETOP framework works with different models as well. We show these results in this section.

## G.1  RESULTS WITH GEMMA

### G.1.1  GEMMA RESULTS ON STRUCTURAL FINE-TUNING

Our results for GEMMA-TT are shown in Table 17, and show that GEMMA-TT is also able to perform well on unseen structural tasks.

Table 17: Results of structural tasks for GEMMA (Zero-shot evaluation), GEMMA-TT (GEMMA fine-tuned with the TREETOP framework). The results show that the fine-tuning process of TREETOP works for smaller LLM models like GEMMA as well.

| Fine-tuning Tasks | GEMMA (Zero-shot) | | | | GEMMA-TT | | | |
|---|---|---|---|---|---|---|---|---|
| | Acc | Rec | Pre | F1 | Acc | Rec | Pre | F1 |
| Are_one_hop_neighbors | 26.3 | 44.4 | 32.6 | 37.6 | 100 | 100 | 100 | 100 |
| Are_two_hop_neighbors | 25.9 | 41.9 | 31.8 | 36.1 | 100 | 100 | 100 | 100 |
| Are_three_hop_neighbors | 23.4 | 41.2 | 30.4 | 35.0 | 100 | 100 | 100 | 100 |
| Within_subtree | 28.2 | 48.7 | 34.4 | 40.3 | **98.0** | 98.0 | 98.0 | 98.0 |
| In_triangle | 23.3 | 43.0 | 31.1 | 36.1 | **86.4** | 99.3 | 79.1 | 88 |
| In_long_chain | 26.0 | 47.7 | 33.0 | 39.0 | **92.7** | 98.7 | 88.2 | 93.1 |
| Num_conversations | 19.6 | 29.3 | 24.4 | 26.7 | 99.0 | 100 | 98.0 | 99.0 |
| Num_children | 18.6 | 23.4 | 21.4 | 22.4 | 99.3 | 99.3 | 99.3 | 99.3 |
| Node_level | 21.6 | 37.3 | 28.3. | 32.2 | **98.7** | 98.7 | 98.7 | 98.7 |
| Is_leaf | 17.7 | 14.0 | 15.1 | 14.5 | 100 | 100 | 100 | 100 |
| Depth | 6.3 | 12.7 | 11.2 | 11.9 | **96.3** | 98.0 | 94.8 | 96.4 |
| Num_leaf_nodes | 22.0 | 33.8 | 27.1 | 30.1 | **78.0** | 87.2 | 73.4 | 79.8 |
| **Unseen Tasks** | | | | | | | | |
| Same_level | 13.4 | 14.9 | 14.5 | 14.7 | **42.4** | 23.2 | 37.7 | 28.7 |
| Has_comment_wedge | 27.5 | 47.9 | 34 | 39.8 | **47.7** | 51.2 | 47.8 | 49.5 |
| Are_interacting | 22.1 | 40.9 | 29.7 | 34.4 | **80.8** | 78.3 | 82.4 | 80.3 |
| Are_fighting | 20.9 | 20.5 | 20.9 | 20.7 | **70.5** | 95.4 | 63.8 | 76.5 |
| Has_user_wedge | 34.7 | 60.0 | 39.8 | 47.9 | **49.5** | 57.3 | 49.5 | 53.2 |

### G.1.2  GEMMA RESULTS ON DOWNSTREAM SOCIAL MEDIA TASKS

Table 18 shows the results of using GEMMA as the base model for the four different social media tasks evaluated in this paper.

Table 18: GEMMA results on downstream social media tasks. These results show that the TREETOP framework helps smaller models like GEMMA as well to improve their downstream performance.

| Task | GEMMA (Zero-shot) | | | | GEMMA (Fine-tuned) | | | | GEMMA-TT (Fine-tuned) | | | |
|---|---|---|---|---|---|---|---|---|---|---|---|---|
| | Acc | Rec | Pre | F1 | Acc | Rec | Pre | F1 | Acc | Rec | Pre | F1 |
| Controversial Post Detection | 17.6 | 31.7 | 24.7 | 27.8 | 64.4 | 96.8 | 58.7 | 73.1 | **80.0** | 79.5 | 80.2 | 79.9 |
| Winning Argument Thread Detection | 8.3 | 4.5 | 4.9 | 4.7 | 50.3 | 98.3 | 50.2 | 66.5 | **71.0** | 79.9 | 66.6 | 72.7 |
| Rumor Detection | 9.5 | 4.0 | 2.4 | 3.0 | 64.9 | 1.6 | 57.1 | 3.1 | **81.8** | 87.9 | 69.0 | 77.3 |
| Fake News Detection | 22.0 | 33.5 | 10.9 | 16.5 | 84.0 | 76.0 | 62.4 | 68.5 | **90.5** | 63.4 | 93.1 | 75.4 |

## G.2 RESULTS WITH PALM-BISON

Our results for PaLM-TT are shown in Table 19, and confirm that the increased benefit of TREETOP framework is repeatable for PaLM-Bison as well.

Table 19: Results of structural tasks for PaLM-Bison (Zero-shot evaluation), PaLM-TT (PaLM-Bison fine-tuned with the TREETOP framework). The results show that the fine-tuning process of TREETOP works for other LLM models like PaLM-Bison as well.

| Fine-tuning Tasks | PaLM (Zero-shot) | | | | PaLM-TT | | | |
|---|---|---|---|---|---|---|---|---|
| | Acc | Rec | Pre | F1 | Acc | Rec | Pre | F1 |
| Are_one_hop_neighbors | 59.5 | 92.1 | 55.8 | 69.5 | **99.9** | 99.7 | 100.0 | 99.9 |
| Are_two_hop_neighbors | 45.6 | 90.1 | 47.7 | 62.4 | **99.3** | 99.1 | 99.6 | 99.3 |
| Are_three_hop_neighbors | 58.1 | 99.5 | 54.4 | 70.3 | **99.6** | 99.5 | 99.7 | 99.6 |
| Within_subtree | 87.2 | 92.0 | 83.9 | 87.7 | **99.6** | 99.3 | 99.9 | 99.6 |
| In_triangle | 61.4 | 72.6 | 59.3 | 65.3 | **93.9** | 92.4 | 95.2 | 93.8 |
| In_long_chain | 76.9 | 89.3 | 71.5 | 79.5 | **98.0** | 98.5 | 97.5 | 98.0 |
| Num_conversations | 61.1 | 98.5 | 56.4 | 71.7 | **99.0** | 98.3 | 99.7 | 99.0 |
| Num_children | 51.6 | 99.2 | 50.8 | 67.2 | **99.5** | 99.1 | 99.9 | 99.5 |
| Node_level | 57.2 | 93.1 | 54.2 | 68.5 | **92.5** | 88.8 | 95.8 | 92.2 |
| Is_leaf | 63.6 | 75.9 | 60.9 | 67.6 | **99.6** | 99.3 | 99.9 | 99.6 |
| Depth | 68.2 | 59.8 | 72.0 | 65.3 | **86.4** | 72.8 | 100.0 | 84.3 |
| Num_leaf_nodes | 50.7 | 24.9 | 51.4 | 33.5 | **90.1** | 96.5 | 85.5 | 90.7 |
| **Unseen Tasks** | | | | | | | | |
| Same_level | 53.0 | 10.3 | 70.8 | 18.0 | **65.3** | 64.8 | 65.5 | 65.1 |
| Has_comment_wedge | 54.9 | 62.2 | 54.2 | 57.9 | **58.1** | 93.6 | 54.8 | 69.1 |
| Are_interacting | 72.9 | 70.8 | 73.9 | 72.3 | **82.2** | 67.4 | 95.7 | 79.1 |
| Are_fighting | 59.3 | 98.5 | 55.2 | 70.7 | **93.7** | 97.9 | 90.3 | 93.9 |
| Has_user_wedge | 54.5 | 49.7 | 55.0 | 52.2 | **59.1** | 51.4 | 60.7 | 55.7 |

## H ENCODING ABLATIONS

In this section, we report ablation results for two different encodings of the post-comment tree: PC, and PCT. These two encodings are as follows:

1. Post Comments (PC): This encoding is designed to capture *only* the content of the conversation tree, *without* the tree structure. In the PC encoding, each comment `encoding`(comment) is encoded as ⟨Content⟩. None of the comment ids, or parent ids are captured.

2. Post Comments Tree (PCT): This encoding captures the content of both the main post and comments along with the structure of the comments tree for that post, and is the one described in Section 3.1. Here, each element in the encoding of the conversation tree has all the richness that we can provide from the dataset.

The prompts, with the prefix, question statement, and conversation tree encoding, is shown in Table 20.

Table 20: This table represents the prompt setup when `Post Comments` and `Conversation Tree` encodings are used to prompt TREETOP. {question} is the task posed as a Yes|No question. If some additional explanation of the question is required, it is added in the {Explanation} section. For example, we first define what a triangular discussion is in a conversation tree and then ask the question about it. Detailed prompts are provide in Section C.3.

| Post with Comments (PC) | Post Conversation Tree (PCT) |
|---|---|
| Given is a social media post and all the comments to the post. Each comment is enclosed in parenthesis i.e (). 

 {Explanation} 

 QUESTION: {question} 

 OPTIONS: 
 – Yes 
 – No 

 POST: {Post content} 

 COMMENTS: (content of comment 1) ... (content of comment n) | Given is a social media conversation tree, where each comment (node) in the tree is of the following structure: (<node_id> <user_id> <content> <parent_id>). The first node (<C0>) is the main post on Reddit followed by the comments to the main post. <parent_id> refers to the comment/post to which the current comment is a reply to. 

 {Explanation} 

 QUESTION: {question} 

 OPTIONS: 
 – Yes 
 – No 

 POST-COMMENT TREE: ⟨encoding(comment$_0$)⟩... |

We compare the performance of GEMINI on both PC and PCT encodings, and compare it with GEMINI-TT (i.e. GEMINI structurally fine-tuned with the TREETOP framework – note that GEMINI-TT uses the PCT encoding). We report results for all the downstream social media tasks (except winning argument detection - which requires the whole conversation tree in the task definition). We employ these comparisons to help us understand the value-added by the tree's topological structure, as LLMs such as GEMINI have inherent language understanding.

Table 21 shows the performance on the controversial post detection task varying the different encodings for the native GEMINI model. Interestingly, we do not see this boost when using the PCT encoding for GEMINI as compared to the PC encoding (accuracy of 68.6% versus 69.2 respectively

for controversial post detection). We posit that this is because GEMINI has no inherent understanding of topology. However, the corresponding accuracy of GEMINI-TT is much higher, at **These results strongly motivate the need for our** TREETOP **framework**: they show that native LLMs like GEMINI are not capable by default to exploit the topological structure provided in the input data.

Table 21: Results for downstream social media tasks for different encodings using the GEMINI model, compared with the GEMINI-TT model. The highest accuracy numbers are in bold, and the best performing model is underlined.

| Controversial Post Detection | | | | | | | |
| --- | --- | --- | --- | --- | --- | --- | --- |
| Model | Zero-shot | | | | Fine-tuned | | |
| | Acc | Rec | Pre | F1 | Acc | Rec | Pre | F1 |
| GEMINI-PC | 53.9 | 28.3 | 61.9 | 38.9 | 69.2 | 85.7 | 65.0 | 73.9 |
| GEMINI-PCT | 50.0 | 27.4 | 58.2 | 37.3 | 68.6 | 85.9 | 64.0 | 73.3 |
| GEMINI-TT | 50.6 | 90.5 | 50.4 | 64.7 | **84.6** | 87.7 | 82.6 | 85.0 |

| Fake News Detection | | | | | | | |
| --- | --- | --- | --- | --- | --- | --- | --- |
| Model | Zero-shot | | | | Fine-tuned | | |
| | Acc | Rec | Pre | F1 | Acc | Rec | Pre | F1 |
| GEMINI-PC | 71.8 | 25.0 | 34.2 | 28.9 | 90.3 | 85.5 | 75.5 | 80.2 |
| GEMINI-PCT | 76.3 | 3.9 | 34.9 | 7.1 | 89.8 | 82.7 | 75.1 | 78.7 |
| GEMINI-TT | 77.2 | 2.8 | 56.8 | 5.3 | **96.0** | 88.6 | 93.5 | 91.0 |

| Rumor Detection | | | | | | | |
| --- | --- | --- | --- | --- | --- | --- | --- |
| Model | Zero-shot | | | | Fine-tuned | | |
| | Acc | Rec | Pre | F1 | Acc | Rec | Pre | F1 |
| GEMINI-PC | 64.8 | 23.9 | 49.5 | 32.2 | 76.3 | 77.6 | 63.3 | 69.7 |
| GEMINI-PCT | 64.0 | 6.1 | 41.3 | 10.6 | 72.5 | 47.1 | 63.7 | 54.1 |
| GEMINI-TT | 65.9 | 3.1 | 76.5 | 5.8 | **87.1** | 78.1 | 83.6 | 80.7 |

# I  DATA ABLATIONS

We perform multiple data ablations, cross-validation, and bootstrapping tests to ensure that the TREETOP framework is stable and our results are not dependent on (i) a particular selection of fine-tuning data corpus, or (ii) a particular train / validation split of our data corpus, (iii) or a particular sampling of the test set. We provide details of these ablation studies here.

## I.1  INCREASING THE SIZE OF THE STRUCTURAL TASK DATASET

### I.1.1  RESULTS ON STRUCTURAL TASKS

For our first data ablation study, we increase the size of the training dataset. Instead of using $10K$ samples per question as described in the previous results, we trained a version of GEMINI-TT with $100K$ samples per task. Like before, half of these were positive samples and the other half, negative samples. We refer to this version of GEMINI-TT as GEMINI-TT-100 and show the comparison results in Table 22.

Table 22: Results of structural tasks for GEMINI-TT and GEMINI-TT-100. Acc, Rec, Pre refer to Accuracy, Recall and Precision respectively. Standard error for all reported results is less than 1.3 for model accuracy for GEMINI-TT-100 and is reported in Appendix I.2 (Table 24). The results show that there is a small gain in performance of GEMINI-TT-100 over GEMINI-TT. The results of GEMINI-TT are same as shown in Table 2, and only presented for comparison purposes.

| Fine-tuning Tasks | GEMINI-TT | | | | GEMINI-TT-100 | | | |
|---|---|---|---|---|---|---|---|---|
| | Acc | Rec | Pre | F1 | Acc | Rec | Pre | F1 |
| Are_one_hop_neighbors | 100.0 | 100.0 | 100.0 | 100.0 | 100.0 | 100.0 | 100.0 | 100.0 |
| Are_two_hop_neighbors | 99.9 | 99.8 | 100.0 | 99.9 | 100.0 | 100.0 | 100.0 | 100.0 |
| Are_three_hop_neighbors | 100.0 | 100.0 | 100.0 | 100.0 | 100.0 | 100.0 | 100.0 | 100.0 |
| Within_subtree | 100.0 | 100.0 | 100.0 | 100.0 | 100.0 | 100.0 | 100.0 | 100.0 |
| In_triangle | 91.6 | 92.1 | 91.1 | 91.6 | **95.4** | 94.1 | 96.6 | 95.3 |
| In_long_chain | 96.5 | 98.7 | 94.6 | 96.6 | **99.7** | 99.7 | 99.6 | 99.7 |
| Num_conversations | 99.9 | 99.9 | 100.0 | 99.9 | 100.0 | 100.0 | 100.0 | 100.0 |
| Num_children | 99.8 | 100.0 | 99.6 | 99.8 | 100.0 | 100.0 | 100.0 | 100.0 |
| Node_level | 94.5 | 99.2 | 90.7 | 94.8 | **100.0** | 100.0 | 100.0 | 100.0 |
| Is_leaf | 99.9 | 99.9 | 100.0 | 99.9 | 100.0 | 100.0 | 100.0 | 100.0 |
| Depth | 93.7 | 90.7 | 96.6 | 93.5 | **100.0** | 100.0 | 100.0 | 100.0 |
| Num_leaf_nodes | 87.7 | 96.7 | 81.9 | 88.7 | **99.6** | 99.6 | 99.6 | 99.6 |
| **Unseen Tasks** | | | | | | | | |
| Same_level | 76.1 | 81.1 | 73.6 | 77.1 | **78.9** | 85.0 | 75.7 | 80.0 |
| Has_comment_wedge | 63.0 | 78.8 | 59.8 | 68.0 | **67.4** | 87.7 | 62.3 | 72.8 |
| Are_interacting | **78.2** | 59.1 | 95.7 | 73.0 | 75.8 | 52.6 | 98.6 | 68.6 |
| Are_fighting | 86.2 | 99.9 | 78.4 | 87.9 | **92.7** | 99.7 | 87.3 | 93.1 |
| Has_user_wedge | 61.6 | 43.3 | 68.3 | 53.0 | **63.9** | 46.3 | 71.6 | 56.2 |

### I.1.2  RESULTS ON DOWNSTREAM SOCIAL MEDIA TASKS

We now show the performance of GEMINI-TT-100 (i.e. GEMINI-TT trained with 100K samples per task) on downstream social media tasks. Table 23 shows the results, and like before, GEMINI-TT-100 consistently shows small gains of GEMINI-TT.

Table 23: Results comparing GEMINI, GEMINI-TT, and GEMINI-TT-100 for downstream social media tasks. All results are statistically significant and the standard error is reported in Appendix I.2. Results for GEMINI and GEMINI-TT are same as before; and are only reported for comparison purposes.

**Controversial Post Detection**

| Model | Zero-shot | | | | Fine-tuned | | | |
|---|---|---|---|---|---|---|---|---|
| | Acc | Rec | Pre | F1 | Acc | Rec | Pre | F1 |
| GEMINI | 50.0 | 27.4 | 58.2 | 37.3 | 68.6 | 85.9 | 64.0 | 73.3 |
| GEMINI-TT | 50.6 | 90.5 | 50.4 | 64.7 | **84.6** | 87.7 | 82.6 | 85.0 |
| GEMINI-TT-100 | 52.5 | 70.0 | 51.9 | 59.6 | **84.5** | 87.7 | 82.6 | 85.1 |

**Winning Argument Thread Detection**

| Model | Zero-shot | | | | Fine-tuned | | | |
|---|---|---|---|---|---|---|---|---|
| | Acc | Rec | Pre | F1 | Acc | Rec | Pre | F1 |
| GEMINI | 50.2 | 3.2 | 51.3 | 6.1 | 51.5 | 12.3 | 57.5 | 20.3 |
| GEMINI-TT | 52.5 | 45.2 | 53.0 | 48.8 | **76.6** | 83.5 | 73.4 | 78.1 |
| GEMINI-TT-100 | 54.6 | 31.4 | 58.9 | 40.9 | **75.5** | 82.7 | 72.4 | 77.2 |

**Fake News Detection**

| Model | Zero-shot | | | | Fine-tuned | | | |
|---|---|---|---|---|---|---|---|---|
| | Acc | Rec | Pre | F1 | Acc | Rec | Pre | F1 |
| GEMINI | 76.3 | 3.9 | 34.9 | 7.1 | 89.8 | 82.7 | 75.1 | 78.7 |
| GEMINI-TT | 77.2 | 2.8 | 56.8 | 5.3 | **96.0** | 88.6 | 93.5 | 91.0 |
| GEMINI-TT-100 | 76.9 | 1.4 | 41.8 | 2.7 | **96.7** | 91.5 | 93.8 | 92.6 |

**Rumor Detection**

| Model | Zero-shot | | | | Fine-tuned | | | |
|---|---|---|---|---|---|---|---|---|
| | Acc | Rec | Pre | F1 | Acc | Rec | Pre | F1 |
| GEMINI | 64.0 | 6.1 | 41.3 | 10.6 | 72.5 | 47.1 | 63.7 | 54.1 |
| GEMINI-TT | 65.9 | 3.1 | 76.5 | 5.8 | **87.1** | 78.1 | 83.6 | 80.7 |
| GEMINI-TT-100 | 65.7 | 2.0 | 100.0 | 3.9 | **87.0** | 79.3 | 83.1 | 81.1 |

Figure 8: We perform bootstrapping analysis on the test set (Efron & Tibshirani, 1993) to ensure that our reported test results are statistically reliable.

## I.2 BOOTSTRAPPING ANALYSIS OF TEST SET

In our next ablation study, we perform bootstrapping of the test set and evaluate against a single version of GEMINI-TT. Bootstrapping is performed using random sampling with replacement within the test data corpus. The methodology and statistical analysis for bootstrapping is described in Efron & Tibshirani (1993) - the process is visualized in Figure 8 and our implementation is given in Algorithm 1. When performing bootstrapping test analysis for structural fine-tuning, the 'model' in Figure 8 refers to GEMINI-TT. However, when performing bootstrapping test analysis for downstream social media tasks, the 'model' in Figure 8 refers to the version of GEMINI-TT-FT specifically fine-tuned for that downstream task. We perform bootstrapping to ensure proper estimation of our test set metrics (accuracy, recall, precision and f1).

---

**Algorithm 1** Estimate Standard Error (SE) Bars via Bootstrap

---

1: **Input:** Test data $T = \{x_1, \ldots, x_n\}$, Number of bootstrap samples $B$, Metric $M : T \to \mathbb{R}$
2: **Output:** SE of the metric evaluated at $T$
3: **for** $i = 1$ to $B$ **do**
4:     Sample $n$ data points from $T$ with replacement to form $S_i$ where $|S_i| = n$
5:     Compute $M_i = M(S_i)$.
6: **end for**
7: Compute the metric for the original population $M^* = M(T)$
8: Compute Standard Error as SE $= \sqrt{\frac{1}{B-1} \sum_{i=1}^{B} (M_i - M^*)^2}$
9: **Return** SE

---

### I.2.1 BOOTSTRAPPING TESTS FOR STRUCTURAL FINE-TUNING

We first show the standard error for structural fine-tuning tasks in Table 24 (for 'mean', refer to Table 2).

Table 24: Mean and standard error from the bootstrapping analysis for structural fine-tuning tasks. The table demonstrates that all standard errors for models in this work are less than 2.0 for accuracy.

Bootstrapping Test Set: Standard Error for Structural Tasks (↓ is better)

| Fine-tuning Tasks | GEMINI (Zero-shot) | | | | GEMINI-TT | | | | GEMINI-TT-100 | | | |
|---|---|---|---|---|---|---|---|---|---|---|---|---|
| | Acc | Rec | Pre | F1 | Acc | Rec | Pre | F1 | Acc | Rec | Pre | F1 |
| Are_one_hop_neighbors | 1.3 | 1.7 | 2.1 | 1.7 | 0.0 | 0.0 | 0.0 | 0.0 | 0.0 | 0.0 | 0.0 | 0.0 |
| Are_two_hop_neighbors | 1.2 | 1.8 | 2.2 | 1.7 | 0.1 | 0.2 | 0.0 | 0.1 | 0.0 | 0.0 | 0.0 | 0.0 |
| Are_three_hop_neighbors | 1.3 | 1.7 | 2.4 | 1.8 | 0.0 | 0.0 | 0.0 | 0.0 | 0.0 | 0.0 | 0.0 | 0.0 |
| Within_subtree | 1.0 | 1.3 | 1.5 | 1.1 | 0.0 | 0.0 | 0.0 | 0.0 | 0.0 | 0.0 | 0.0 | 0.0 |
| In_triangle | 2.0 | 2.7 | 3.3 | 2.7 | 0.7 | 1.0 | 1.0 | 0.7 | 0.3 | 0.6 | 0.4 | 0.3 |
| In_long_chain | 1.8 | 1.1 | 14.7 | 2.0 | 0.4 | 0.4 | 0.7 | 0.4 | 0.0 | 0.1 | 0.1 | 0.0 |
| Num_conversations | 1.5 | 10.2 | 1.8 | 5.1 | 0.1 | 0.1 | 0.0 | 0.1 | 0.0 | 0.0 | 0.0 | 0.0 |
| Num_children | 1.5 | 2.0 | 2.7 | 2.1 | 0.1 | 0.0 | 0.2 | 0.1 | 0.0 | 0.0 | 0.0 | 0.0 |
| Node_level | 1.2 | 1.4 | 1.5 | 1.2 | 0.0 | 0.0 | 0.0 | 0.0 | 0.0 | 0.0 | 0.0 | 0.0 |
| Is_leaf | 1.9 | 3.1 | 3.5 | 3.0 | 0.1 | 0.1 | 0.0 | 0.1 | 0.0 | 0.0 | 0.0 | 0.0 |
| Depth | 1.3 | 1.4 | 3.4 | 1.9 | 0.1 | 0.2 | 0.2 | 0.1 | 0.0 | 0.0 | 0.0 | 0.0 |
| Num_leaf_nodes | 1.3 | 2.0 | 2.1 | 1.7 | 1.0 | 0.7 | 1.4 | 0.9 | 0.1 | 0.1 | 0.1 | 0.1 |
| **Unseen Tasks** | | | | | | | | | | | | |
| Same_level | 1.3 | 0.9 | 5.9 | 1.6 | 1.2 | 1.6 | 1.6 | 1.3 | 1.2 | 1.4 | 1.6 | 1.2 |
| Has_comment_wedge | 1.5 | 1.1 | 3.9 | 1.7 | 1.2 | 1.5 | 1.4 | 1.3 | 1.3 | 1.3 | 1.5 | 1.2 |
| Are_interacting | 1.2 | 0.8 | 5.3 | 1.4 | 1.1 | 1.8 | 0.9 | 1.4 | 1.1 | 1.9 | 0.6 | 1.6 |
| Are_fighting | 1.2 | 1.4 | 3.1 | 2.0 | 0.9 | 0.1 | 1.3 | 0.8 | 0.7 | 0.2 | 1.1 | 0.7 |
| Has_user_wedge | 1.4 | 1.2 | 3.9 | 1.8 | 1.3 | 1.9 | 2.1 | 1.8 | 1.2 | 1.9 | 2.1 | 1.6 |

### I.2.2 BOOTSTRAPPING TESTS FOR DOWNSTREAM SOCIAL MEDIA TASKS

We now report the standard error for the controversial post detection task in Table 25 (for 'mean', refer to Table 3). Similarly, we report standard error for winning argument thread detection task in Table 26 (for 'mean', refer to Table 4), for fake news detection in Table 27 (for 'mean', refer to Table 15), and for rumor detection in Table 28 (for 'mean', refer to Table 16).

Table 25: Standard error for the controversial post detection task. The table demonstrates that all standard errors for model accuracy in this evaluation are less than 1.3.

Bootstrapping Test for Controversial Post Detection
Standard Error (↓ is better)

| Model | Zero-shot | | | | Fine-tuned | | | |
|---|---|---|---|---|---|---|---|---|
| | Acc | Rec | Pre | F1 | Acc | Rec | Pre | F1 |
| GEMINI | 1.3 | 1.6 | 2.4 | 1.8 | 1.0 | 1.0 | 1.3 | 1.0 |
| GEMINI-TT | 1.0 | 0.9 | 1.1 | 1.0 | 0.8 | 1.0 | 1.1 | 0.8 |
| GEMINI-TT-100 | 1.1 | 1.5 | 1.3 | 1.2 | 0.7 | 0.9 | 1.1 | 0.7 |

Table 26: Standard error for the winning argument thread detection task. The table demonstrates that all standard errors for model accuracy in this evaluation are less than 1.4.

Bootstrapping Test for Winning Argument Thread Detection Task
Standard Error (↓ is better)

| Model | Zero-shot | | | | Fine-tuned | | | |
|---|---|---|---|---|---|---|---|---|
| | Acc | Rec | Pre | F1 | Acc | Rec | Pre | F1 |
| GEMINI | 1.4 | 0.7 | 7.6 | 1.2 | 1.1 | 1.1 | 3.2 | 1.6 |
| GEMINI-TT | 1.1 | 1.6 | 1.7 | 1.4 | 0.9 | 1.2 | 1.3 | 1.0 |
| GEMINI-TT-100 | 1.0 | 1.5 | 1.7 | 1.5 | 1.0 | 1.2 | 1.3 | 1.0 |

Table 27: Standard error for the fake news detection task. The table demonstrates that all standard errors for model accuracy in this evaluation are less than 0.2.

Bootstrapping Test for Fake News Detection Task
Standard Error (↓ is better)

| Model | Zero-shot | | | | Fine-tuned | | | |
|---|---|---|---|---|---|---|---|---|
| | Acc | Rec | Pre | F1 | Acc | Rec | Pre | F1 |
| GEMINI | 0.2 | 0.2 | 1.7 | 0.4 | 0.2 | 0.4 | 0.4 | 0.3 |
| GEMINI-TT | 0.2 | 0.2 | 2.3 | 0.3 | 0.1 | 0.4 | 0.3 | 0.3 |
| GEMINI-TT-100 | 0.2 | 0.1 | 2.8 | 0.2 | 0.1 | 0.3 | 0.3 | 0.2 |

Table 28: Standard error for the rumor detection task. The table demonstrates that all standard errors for model accuracy in this evaluation are less than 1.9.

Bootstrapping Test for Rumor Detection Task
Standard Error (↓ is better)

| Model | Zero-shot | | | | Fine-tuned | | | |
|---|---|---|---|---|---|---|---|---|
| | Acc | Rec | Pre | F1 | Acc | Rec | Pre | F1 |
| GEMINI | 1.5 | 1.8 | 9.5 | 3.0 | 1.9 | 3.9 | 4.5 | 3.7 |
| GEMINI-TT | 1.8 | 1.2 | 15.1 | 2.3 | 1.3 | 2.6 | 2.6 | 2.1 |
| GEMINI-TT-100 | 1.8 | 0.9 | 0.0 | 1.8 | 1.2 | 2.4 | 2.8 | 2.0 |

## I.3 TREETOP FRAMEWORK FINE-TUNING CORPUS ABLATIONS

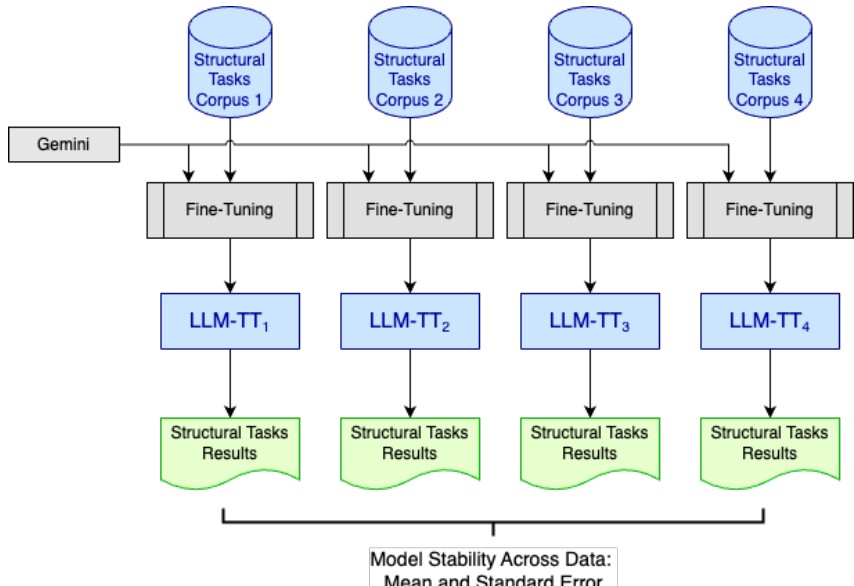

Figure 9: We change the entire structural task fine-tuning corpus and train multiple versions of GEMINI-TT to ensure that our results are not sensitive to any particular choice of structural task data corpus, and that our methodology of fine-tuning using the structural tasks is repeatable.

In the second ablation study, we change the entire corpus of the structural task data set multiple times to create multiple versions of GEMINI-TT. Our process is described in Figure 9. We perform this ablation study ensure that our process of fine-tuning using a structural task dataset is repeatable and not dependent on just one specific selection of data set. In our study, we created four independent non-overlapping dumps of the dataset, and we report the mean and standard error across this data dump ablation in Table 29.

Table 29: Standard error across four data dump ablation study for GEMINI-TT. As the table shows, the standard errors are less than 1.54 for accuracy.

| | Mean ($\uparrow$ is better) | | | | Standard Error ($\downarrow$ is better) | | | |
|---|---|---|---|---|---|---|---|---|
| **Fine-tuning Tasks** | **Acc** | **Rec** | **Pre** | **F1** | **Acc** | **Rec** | **Pre** | **F1** |
| Are_one_hop_neighbors | 100.00 | 100.00 | 100.00 | 100.00 | 0.00 | 0.00 | 0.00 | 0.00 |
| Are_two_hop_neighbors | 99.78 | 99.73 | 99.83 | 99.78 | 0.1 | 0.07 | 0.11 | 0.1 |
| Are_three_hop_neighbors | 99.95 | 99.95 | 100.00 | 99.95 | 0.03 | 0.03 | 0.00 | 0.03 |
| Within_subtree | 100.00 | 100.00 | 100.00 | 100.00 | 0.00 | 0.00 | 0.00 | 0.00 |
| In_triangle | 89.13 | 91.43 | 87.48 | 89.38 | 0.8 | 0.40 | 1.36 | 0.76 |
| In_long_chain | 96.80 | 98.90 | 94.93 | 96.90 | 0.24 | 0.16 | 0.49 | 0.24 |
| Num_conversations | 99.93 | 99.90 | 99.98 | 99.93 | 0.03 | 0.07 | 0.03 | 0.03 |
| Num_children | 99.65 | 99.95 | 99.33 | 99.65 | 0.09 | 0.03 | 0.16 | 0.09 |
| Node_level | 95.68 | 99.08 | 92.98 | 95.90 | 1.47 | 0.35 | 2.39 | 1.39 |
| Is_leaf | 99.98 | 99.98 | 100.00 | 99.98 | 0.03 | 0.03 | 0.00 | 0.03 |
| Depth | 95.18 | 93.78 | 96.53 | 95.10 | 1.54 | 2.02 | 1.21 | 1.66 |
| Num_leaf_nodes | 86.80 | 94.48 | 81.93 | 87.73 | 0.54 | 1.14 | 0.62 | 0.53 |
| **Unseen Tasks** | Mean | | | | Standard Error | | | |
| Same_level | 75.40 | 77.17 | 74.60 | 75.80 | 0.35 | 1.81 | 0.65 | 0.65 |
| Has_comment_wedge | 61.97 | 76.10 | 59.33 | 66.63 | 0.54 | 1.31 | 0.32 | 0.7 |
| Are_interacting | 78.43 | 59.40 | 95.97 | 73.40 | 0.31 | 1.13 | 0.81 | 1.13 |
| Are_fighting | 86.10 | 99.77 | 78.37 | 87.80 | 0.10 | 0.06 | 0.13 | 0.10 |
| Has_user_wedge | 62.07 | 42.83 | 69.73 | 53.03 | 0.36 | 0.32 | 0.63 | 0.18 |

TREETOP Framework Corpus Ablation Study

## I.4 K-Fold Cross-Validation for Downstream Social Media Tasks

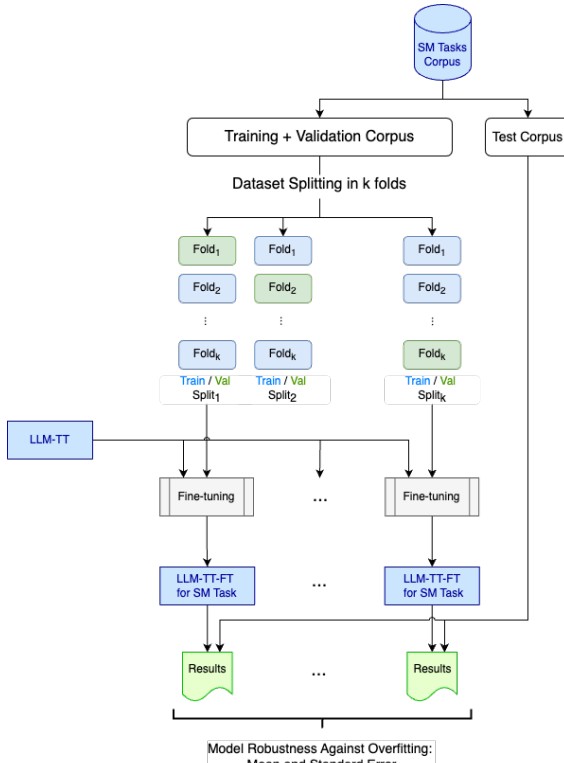

Figure 10: We perform the standard k-fold cross validation for each downstream social media task with GEMINI-TT to ensure that our results are not over-fitted for any particular fold of the dataset.

In our third ablation study, we perform standard k-fold cross-validation for each downstream task. We create multiple versions of fined-tuned GEMINI-TT for each downstream task. Our process is described in Figure 10. We perform this ablation study ensure that our process of fine-tuning using a structural task dataset is not over-fitted to any one particular training data corpus. Our results and standard error are shown in Table 30.

Table 30: Mean and standard error for the k-fold cross-validation study for the different algorithms for GEMINI-TT. The standard errors are less than 0.32 for accuracy.

| Task | Mean (↑ is better) | | | | Standard Error (↓ is better) | | | |
|---|---|---|---|---|---|---|---|---|
| | Acc | Rec | Pre | F1 | Acc | Rec | Pre | F1 |
| Controversial Post Detection | 84.22 | 87.88 | 81.96 | 84.78 | 0.16 | 1.29 | 0.86 | 0.17 |
| Winning Argument Thread Detection | 76.52 | 79.58 | 74.98 | 77.18 | 0.18 | 1.36 | 0.52 | 0.39 |
| Rumor Detection | 86.60 | 79.12 | 82.02 | 80.5 | 0.27 | 0.36 | 0.80 | 0.31 |
| Fake News Detection | 97.14 | 92.88 | 94.6 | 93.72 | 0.32 | 1.25 | 0.49 | 0.76 |

## J  EMBEDDINGS ANALYSIS

We also analyzed the embeddings of PaLM-TT, with an intent to test whether the model actually understood the topology - we designed our experiment to evaluate whether the topological characteristics are directly captured in the learnt embeddings output by the model. We used PaLM-TT for this analysis, given it was not possible for us to obtain embeddings from GEMINI-based models.

Our experimental results show a clear presence of clustering in the embedding space for inputs that conform to similar topologies. In fact, a classifier trained purely on topological embeddings obtains a 77% accuracy on the multi-class topology classification task using a simple logistical regression classifier. We now describe the experimental setup that leads to these conclusions.

### J.1  EXPERIMENTAL SETUP

In this section, we describe our experimental setup. The goal of these experiments is to show that embeddings learnt by the PaLM-TT model show clustering — i.e. topologies that are similar, by some independent analytical metric, cluster together. Towards this goal, we chose 5 different topologies, which are shown in Figure 11. These five topologies are chosen so that they vary in their depth and breadth. The hypothesis at the outset is to show that embeddings cluster by topology.

We first begin by presenting the statistics of our topologies in Table 31. While the topologies are illustrated in Figure 11, we list them out in our table with an equivalent depth-first crawl of the tree: the list of children of a node are captured with a pair of parenthesis, and the letter $v$ represents a leaf.

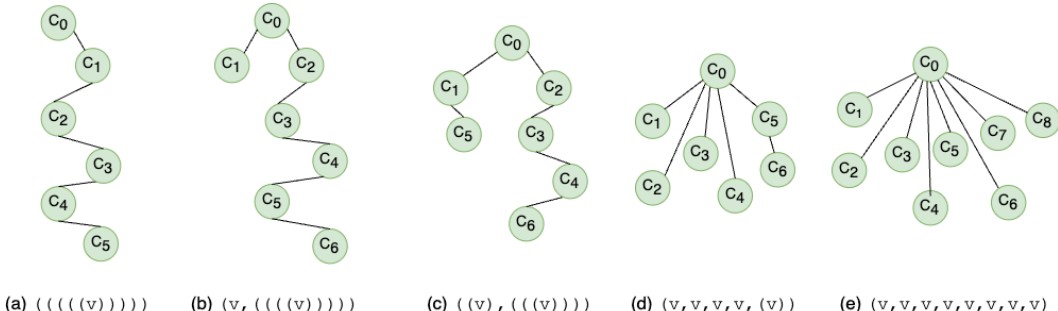

(a) ((((v))))  (b) (v,((((v)))))  (c) ((v),(((v))))  (d) (v,v,v,v,(v))  (e) (v,v,v,v,v,v,v,v)

Figure 11: Five topologies used for PaLM-TT embedding analysis.

Table 31: We randomly chose some topologies to evaluate clustering of embeddings. These topologies vary given the first one has high depth and the last one has high breadth. They are shown in Figure 11.

| Topology | Number of Samples |
|---|---|
| ((((v)))) | 659 |
| (v,((((v))))) | 436 |
| ((v),(((v)))) | 602 |
| (v,v,v,v,(v)) | 1252 |
| (v,v,v,v,v,v,v,v) | 382 |

We now describe the method by which we obtain the embeddings. The embedding is obtained by doing a forward pass of the PCT encoding in PaLM-TT for each conversation tree in our dataset. We believe that the *last input token's* embedding is sufficient capture the entire prompt at inference time, and so we only use the embedding of the last input token in our analysis. Let's call such an embedding as E(PCT). Similarly, we forward pass the Post Comments encoding (refer Appendix C.2) from the same conversation tree, and obtain E(PC). The embedding dimension for each token in PaLM-TT (as in PaLM-Bison) is 4096. Now we are left with the task of obtaining embeddings – call this E(Topo) that refer purely to the embeddings of the topology. We define

this as

$$E(\text{Topo}) = E(\text{PCT}) - E(\text{PC}) \tag{1}$$

Our assumption is that `E(Topo)` is a good approximation for the topological structure, and we show our analysis on these `E(Topo)` embeddings. Our hypothesis is that `E(Topo)` will demonstrate correlation with the actual tree topology.

To demonstrate that our embeddings capture patterns from the respective topologies, we examine two types of metrics. First, we aim to show separability between embeddings from different clusters. We hypothesize that if a linear (or logistic) classifier can effectively learn the classes using the embeddings, this would demonstrate such separability. Next, we look at a clustering distance metric – called the variance ratio criterion. We hope to show that the `VRC` metrics reflects the distances of another independent metric, for example the tree edit-distance metric as proposed by the `Zhang Shasha` algorithm (Zhang & Shasha, 1989). We describe these experiments in the next two sections.

### J.2 MULTI-CLASS TOPOLOGICAL CLASSIFIER USING PaLM-TT'S EMBEDDINGS

In the first study, we utilized embeddings to classify various topological classes, achieving impressive performance metrics as detailed in Table 32. We studied two different types of models to explain the complexity of the dataset – (i) logistical classifier (ii) a two-layer MLP. If the data cannot be learnt by the first classifier, but can be learnt by the second, it would show that the data has complex patterns. If it can be learnt by both, then the embeddings show clean separation between the different classes. If it cannot be learnt by either model, then the embeddings may not capture the topological patterns (or otherwise the patterns are too complex to be learnt by these simple models).

Table 32: Performance metrics for Neural Network and Logistic Regression models on the topological embedding to classify the topological class.

| Neural Network | | | | |
|---|---|---|---|---|
| **Topology** | **Accuracy** | **Precision** | **Recall** | **F1** |
| `((((( v )))))` | 73.0 | 71.0 | 72.0 | |
| `(v,((((v)))))` | 40.0 | 31.0 | 35.0 | |
| `((v),(((v))))` | 55.0 | 62.0 | 58.0 | |
| `(v,v,v,v,(v))` | 84.0 | 89.0 | 87.0 | |
| `(v,v,v,v,v,v,v,v)` | 74.0 | 67.0 | 70.0 | |
| **Overall** | 72.0 | 71.0 | 72.0 | 72.0 |
| **Logistic Regression** | | | | |
| `((((( v )))))` | 79.0 | 77.0 | 78.0 | |
| `(v,((((v)))))` | 45.0 | 50.0 | 47.0 | |
| `((v),(((v))))` | 65.0 | 62.0 | 63.0 | |
| `(v,v,v,v,(v))` | 90.0 | 90.0 | 90.0 | |
| `(v,v,v,v,v,v,v,v)` | 78.0 | 77.0 | 78.0 | |
| **Overall** | **77.0** | **77.0** | **77.0** | **77.0** |

We show the topological classes in Figure 11 and their statistics in Table 31. Note that *both* the models can learn the dataset, to the extent of 77% accuracy! This shows that the embeddings cleanly capture the topological patterns. These metrics underscore the **efficacy of our approach in capturing the characteristics of topological classes** using embeddings.

### J.3 CLUSTERING METRICS FROM THE EMBEDDING

Our topological embeddings show that the tree distance metric is highly correlated with the variance ratio criterion (VRC) metric. The VRC is defined as follows:

$$\text{VRC} = \frac{\text{Tr}(B_k)}{\text{Tr}(W_k)} \times \frac{N - k}{k - 1} \tag{2}$$

where the $\text{Tr}(B_k)$ is the trace of the between-cluster dispersion matrix, $\text{Tr}(W_k)$ is the trace of the within-cluster dispersion matrix, and $N$ and $k$ are the number of data points and the number of clusters respectively (and listed in Table 31).

Here the trace of $B_k$ and $W_k$ are defined as follows:

$$\text{Tr}(B_k) = \sum_{j=1}^{k} n_j (\mathbf{c}_j - \mathbf{c})^2 \quad \text{and} \quad \text{Tr}(W_k) = \sum_{j=1}^{k} \sum_{\mathbf{x} \in C_j} (\mathbf{x} - \mathbf{c}_j)^2 \tag{3}$$

Here, each topology is its own cluster, and $\mathbf{c}_j$ is the centroid of cluster $j$. $\mathbf{c}$ is the centroid of all data points in the dataset. Further, $n_j$ is the number of data points in cluster $j$, and the set of all points in cluster $j$ is $C_j$. $\mathbf{x}$ represents the embedding, `E(Topo)`, for the sample in question.

The tree distance metric is an edit distance between the topologies of the respective trees. This is implemented using the `Zhang Shasha` (Zhang & Shasha, 1989) algorithm. One notices that the VRC metric of the trees in the cluster closely follows the edit distance between the topologies of the clusters.

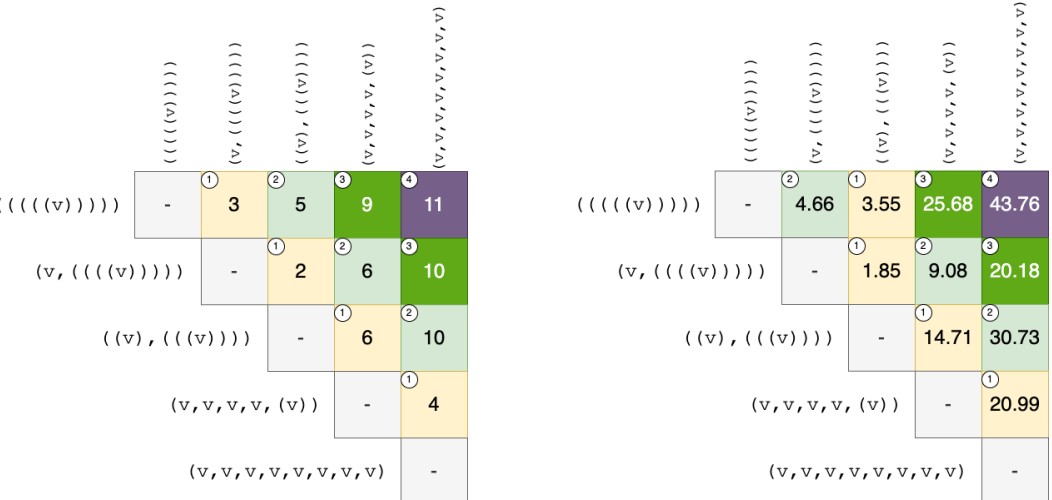

(a) Tree edit distance between our topologies based on the `Zhang Shasha` algorithm.

(b) Variance ratio criterion (VRC) metric for different clusters of `E(Topo)`.

Figure 12: Comparison of PaLM-TT embeddings with VRC criterion and tree edit distance. Each cell shows the value of the pairwise distance metric (self distances are 0 for tree edit distance and not defined for VRC), the ordinal ranking of the cell in the top-left corner, and is color coded by this ordinal rank. The ordinal ranking ranks the topologies in the columns based on their distance from the topology specified in the row. Figure 12a shows the color map with the ordinal rankings based on the tree edit distance measure, whereas Figure 12b shows the color map based on the variance ratio. The comparison is **of the ordinal ranking** and **not of the absolute metric** values. The similar color coding of the two plots implies similar ordinal ranking and demonstrates that VRC metric of the trees in the cluster closely follows the edit distance between the topologies of the clusters - thereby implying that embeddings are indeed learning the topology.

## K  PROMPT INTERACTIONS WITH GEMINI AND GEMINI-TT

We randomly chose few examples to probe GEMINI and GEMINI-TT models for our structural tasks. We present the tasks and the interactions with the models in Tables 33, 34, and 35. For this table, we changed our prompt slightly and asked the model to provide more details (instead of only responding with a `yes` or a `no`). We present three example interactions here: one for a `In_long_chain` task, one for `In_triangle` task, and one for a `Are_three_hop_neighbors` task. In all the three examples below, GEMINI gave the wrong response whereas GEMINI-TT gave the correct response with the correct reasoning.

Table 33: Example Task 1: Interacting with GEMINI and GEMINI-TT with our structural tasks.

| Task | **In_long_chain** |
|------|-------------------|
| Input Topology |  |
| Prompt Question | Whether there exist a long chain of to and fro discussion of at least length 4 between two users in the tree ? |
| GEMINI Response | **No,** there is no long chain of to and fro discussion of at least length 4 between two users.

The conversation tree given only has a maximum chain length of 3, between U0 and U1.  The chain goes as follows:
1.  U0 comments on the main post <C0>
2.  U1 comments on U0's comment <C1>
3.  U0 comments on U1's comment <C2>

There is no instance where U1 comments on U0's comment to U1's comment, or vice versa. Therefore, there is no long chain of to and fro discussion of at least length 4 between two users. |
| GEMINI-TT Response | **Yes,** there is a long chain of to and fro discussion of length 4 between users U0 and U1.  The relevant comments are:
– C1:  U1 comments on a comment by U0 (C0).
– C2:  U0 comments on U1's comment (C1).
– C3:  U1 comments on U0's comment (C2).
– C4:  U0 comments on U1's comment (C3). |

Table 34: Example Task 2: Interacting with GEMINI and GEMINI-TT with our structural tasks.

| **Task** | **Are_three_hop_neighbors** |
|---|---|
| Input Topology |  |
| Prompt Question | Whether <C4> is a three-hop neighbor to <C0> ? |
| GEMINI Response | **No,** <C4> is not a three-hop neighbor to <C0>.

In the given post-comment tree:
- <C0> is the main post.
- <C1>, <C2>, and <C3> are direct replies to <C0>, making them one-hop neighbors to <C0>.
- <C4> is a reply to <C3>, which is two hops away from <C0>.

Therefore, while <C4> is a neighbor of <C0>, it is not a three-hop neighbor because the path from <C0> to <C4> consists of only two hops: <C0> -> <C3> -> <C4>.

users <U0>, <U1>, and <U2> are not involved in a triangular discussion between each other. |
| GEMINI-TT Response | **Yes,** <C4> is a three-hop neighbor to <C0>.

The path from <C0> to <C4> is as follows:
<C0> -> <C2> -> <C3> -> <C4>

The users and comment IDs of the relevant comments are:
- <U0> (<C0>)
- <U2> (<C2>)
- <U0> (<C3>)
- <U2> (<C4>) |

Table 35: Example Task 3: Interacting with GEMINI and GEMINI-TT with our structural tasks.

| Task | **In_triangle** |
|------|----------------|
| Input Topology |  |
| Prompt Question | Whether users <U0>, <U1>, and <U2> are not involved in a triangular discussion between each other.   ? |
| GEMINI Response | **No**, users <U0>, <U1>, and <U2> are not involved in a triangular discussion between each other.

Although users <U1> and <U2> have commented on each other's comments, user <U0> has not commented on either of their comments. Therefore, there is no triangular discussion between the three users. |
| GEMINI-TT Response | **Yes**, users <U0>, <U1> and <U2> are involved in a triangular discussion.
– User <U1> comments on a comment by User <U0> (C1).
– User <U2> comments on the same comment by User <U0> (C2).
– User <U1> and User <U2> comment on each other's comment on User <U0>'s comment (C3).

Therefore, the three users are involved in a triangular discussion. |

## L    LICENSES AND COPYRIGHTS ACROSS ASSETS

1. The Pushshift Reddit Dataset
   - Citation: (Baumgartner et al., 2020)
   - Asset Link: [link]
   - License: CC By 4.0

2. Controversial Post
   - Citation: (Hessel & Lee, 2019)
   - Asset Link: [link]
   - License: CC By 4.0

3. PHEME9 (rumor detection)
   - Citation: (Kochkina et al., 2018)
   - Asset Link: [link]
   - License: CC By 4.0

4. Fakeddit
   - Citation: (Nakamura et al., 2019)
   - Asset Link: [post] [comments]
   - License: CC By 4.0

5. Winning Argument Thread
   - Citation: (Tan et al., 2016)
   - Asset Link: [link]
   - License: ACM Copyright

6. GEMINI
   - Citation: (Team et al., 2023)
   - Asset Link: [link]
   - License: Google APIs Terms of Service

7. PaLM-Bison
   - Citation: (Google and et al., 2023)
   - Asset Link: [link]
   - License: Google APIs Terms of Service

8. Huggingface Assets
   - GEMMA-2-2B-IT: [link]
   - PHI-3.5-Mini-Instruct: [link]
   - MISTRAL-7B-Instruct-v0.2: [link]
   - GEMMA-2-9B-IT: [link]
   - MMLU: [link]
   - GSM8K: [link]

