# OpenReview forum: "TreeTop: Topology-Aware Fine-Tuning for LLM Conversation Tree Understanding"
_ICLR.cc/2025/Conference — Submitted to ICLR 2025_

### Official Review · Reviewer_5ohp · 2024-10-31

**Soundness:** 2
**Presentation:** 2
**Contribution:** 2
**Rating:** 5
**Confidence:** 3

**Summary:**

TreeTop is a fine-tuning framework designed to enhance LLMs in understanding structured inputs like conversation trees on social media. Featuring 17 tasks, TreeTop focuses on improving LLMs' ability to reason about structural and content relationships in these discussions. LLMs fine-tuned with TreeTop outperform baseline models, including SOTA GNNs, in generalizing to new tasks and excelling in social media inference tasks, such as controversy detection, even in early-detection scenarios. This framework advances LLMs' capabilities in processing structured data.

**Strengths:**

1. This work presents a new conversation tree encoding method. It would be beneficial to the field of multi-turn multi-party conversation.

2. It reveals the shortcoming of existing LLMs on understanding such kinds of conversation tree, and also presents a fine-tuning-based solution.

**Weaknesses:**

1. I think the writing should be improved. For example, I did not catch how you finetune LLMs with the TreeTop framework. It is better to clarify it clearly in the main content.

2. It seems that the baselines and related works are out-of-the-date. Are there any other more recently works of LLM on conversation tree?

3. I wonder how the conversation tree benefits the downstream dialogue tasks, such as dialogue response generation. Generating and understanding the conversation tree should be the endpoints of exploring this technique. Constructing the conversation tree, I think, should facilitate the quality of the response generation in multi-party conversations. So, I think it is important to highlight how the conversation tree will help the field of dialogue system.

**Questions:**

See weakness.

---

> ### Author Response · Authors · 2024-11-21
>
> We thank Reviewer 5ohp for their thoughtful feedback and for recognizing both the novelty of our conversation tree encoding method and its potential benefits to multi-turn, multi-party conversations. We also appreciate the reviewer highlighting how our work reveals important limitations in existing LLMs' ability to understand conversation trees. We address each concern below:
>
>
> **Weakness 1:**
>
> **Reviewer Comment:** "I did not catch how you finetune LLMs with the TreeTop framework. It is better to clarify it clearly in the main content."
>
> Response: Thank you for the suggestion. We agree that the primary details relating to fine-tuning should be in the main content. We have added the following sentences to Section 4.1:
>
> "Each model was fine-tuned on 10k randomly-sampled tasks from each of the 12 "fine-tuning tasks" listed in Table 2, holding out the other 5 "unseen" tasks for out-of-distribution evaluation. We adopt the unified encoding presented in Section 3.1. We list hardware and hyperparameter details in Appendix F."
>
> **Weakness 2:**
>
> **Reviewer Comment:** "It seems that the baselines and related works are out-of-the-date. Are there any other more recently works of LLM on conversation tree?"
>
> Response: While we appreciate this concern about comparisons, we wish to highlight that ours is one of the nascent works on providing LLMs with a conversation tree understanding. Most of the recent LLM+graph works focus on traditional graph tasks like edge classification and node classification (see for example Pan et al., 2024; Tsitsulin et al., 2024; Chen et al., 2024b).
>
> For non-LLM related benchmarks, we have compared against the most recent applicable works on our datasets, including  PHAROS for rumor detection (Nguyen et al., 2024) and DFE-GCN for controversy detection (Hua et al., 2023). We note that a key challenge for expanding over datasets comparisons is data availability – publicly available datasets with complete conversation tree structures are increasingly scarce, especially from platforms like Twitter, which now require paid API access. Our work represents one of the first focused efforts to enhance LLMs' capabilities on conversation tree understanding.
>
> **Weakness 3:**
>
> **Reviewer Comment:** "I wonder how the conversation tree benefits the downstream dialogue tasks, such as dialogue response generation."
>
> Response: Thank you for this insightful suggestion about dialogue systems. It's important to note that social media conversations have fundamentally different structural characteristics from traditional two-person dialogues (as with a chatbot). On social media platforms, multiple conversational threads emerge simultaneously in response to the same post, creating complex tree topologies that carry important signals about the discussion's nature (e.g., controversy, rumors). TreeTop specifically aims to exploit these unique structural patterns.
>
> For example, a controversial post might generate multiple divergent reply branches. Similarly, a hateful post may display specific tree patterns -- these structural signals are distinct from traditional dialogue flows. Section 4.2 demonstrates how TreeTop effectively leverages these topology-specific signals, achieving strong performance on tasks like controversy detection (84.6% accuracy) and rumor detection (87.1% accuracy).
>
> Our current framework is specifically designed to capture the unique multi-threaded nature of social media conversations, where topology provides critical additional signals beyond content alone. Nonetheless, we agree that exploring TreeTop's application to traditional dialogue systems is an interesting direction for future work.
>
> We hope our responses adequately address your concerns. If you find these clarifications satisfactory, particularly regarding the fine-tuning methodology and the recency of our baselines, we kindly request you to consider raising your score.

---

> > ### Comment · Reviewer_5ohp · 2024-11-26
> >
> > Thanks for author's reply. I have carefully read the comments from other reviewers and author's response. However, my concerns regarding the novelty of the work, its scope of applicability, and the limited recent relevant studies remain unresolved. Therefore, I maintain my current score.

---

> ### Author Response · Authors · 2024-11-30
>
> Thanks for the reviewer's response.
>
> We believe that our work is novel. By-and-large, the studies that use LLMs for graphs have tackled tasks on undirected, cyclic, static graphs, with applications geared toward standard GNN benchmarks (Hu et al., 2020a). In a parallel line of work, language models have been used to improve GNN performance. We describe these related works in Section 2 and Appendix B.
> Extending this area of research, we focus on fine-tuning LLMs to solve learning tasks on conversation trees, which are directed, acyclic, and temporal graphs capable of representing a wide variety of complex human interaction sequences. Our work opens a new area of research for LLMs, as well as providing the power of LLMs to the field of conversation tree analysis (or to text-attributed graphs broadly).
>
> We have presented experiments on multiple datasets where conversation trees were available, and compared with the most recent results on those datasets. However, we would be happy to provide more empirical evidence and comparisons if the reviewer can point us in the direction of additional relevant papers and datasets.

---

> > ### Author Response · Authors · 2024-12-02
> > **Request for further questions and feedback**
> >
> > Thank you for your initial review and your reply to our rebuttal. We are wondering if our further response clarified matters. If so, we humbly request that you reconsider your score. If not, we are happy to answer any follow-up questions or comments. Cheers,
> >
> > Authors

---

### Official Review · Reviewer_iBfG · 2024-11-01

**Soundness:** 3
**Presentation:** 3
**Contribution:** 2
**Rating:** 5
**Confidence:** 4

**Summary:**

In this paper, the author proposed TreeTop, a fine-tuning framework improving the reasoning capabilities of LLMs over structured conversation tree data. TreeTop incorporates 17 structural tasks to train LLMs on the unique topological and content-driven aspects of conversation trees. Experimental results show that LLMs fine-tuned with TreeTop outperform traditional GNNs and other baselines on social media inference tasks.

**Strengths:**

- Adapting LLMs for graph-structured conversation trees are distinct graph problem-solving methods from traditional, static graphs in their directed, acyclic nature and temporal aspects.
- The evaluation on various social media tasks is comprehensive. Fine-tuning on structural tasks and subsequent performance comparisons are sufficient. The ablation analysis is well-set up and indicates the stability of the proposed method.

**Weaknesses:**

- This framework may not sufficiently address how to perform with longitudinal data, where multiple conversation trees evolve over time or space.
- The framework could benefit from considering the integration of user characteristics across different conversation trees, especially in social media data where the same user may participate in multiple sessions.
- The details of integrating 17 structural tasks to fine-tune LLMs, such as considering the insightful associations and importance weights between these tasks, are not yet fully clarified.
- In terms of comparisons, only traditional GNNs are used as the main baselines, lacking comparisons with more recent graph-based reasoning methods.
- The reliance on task-specific prompts requires more results to clarify the extent to which various prompt styles across tasks influence TreeTop’s effectiveness.

**Questions:**

- How are the 17 structural tasks weighted during training? Is their importance dynamically adjusted based on downstream application requirements?
- Can TreeTop model the associations between multiple conversation trees, especially based on longitudinal data or same user scenarios?
- How does TreeTop differentiate between structure- and content-related inferences when solving a reasoning task?
- How does TreeTop compare to other fine-tuning based methods on general graph-based reasoning tasks, such as edge existence prediction or node classification?
- How sensitive are the model outputs to prompt design, and is there a mechanism to automatically adjust prompts for different tasks?

---

> ### Author Response · Authors · 2024-11-21
>
> Thank you for the worthwhile questions. We respond to each below. We request that you please reconsider our work in light of these responses. If they have sufficiently addressed your concerns, please consider raising your score. Thank you!
>
> **Reviewer Question:** "How are the 17 structural tasks weighted during training? Is their importance dynamically adjusted based on downstream application requirements?”
>
> As described in Section 3.2, we only use 12 of the structural tasks during fine-tuning (i.e. the training that occurs after general pretraining). Each of these tasks are weighted equally -- downstream tasks are not taken into account. We find that this maximizes TreeTop's generalizability to a wide range of downstream tasks.
>
> We chose uniform weighting for simplicity and reproducibility, as exploring differential weighting schemes would introduce significant additional complexity without clear benefits. Our results show that models fine-tuned with these 12 equally-weighted TreeTop structural tasks, and then further fine-tuned on various downstream tasks, consistently outperform models fine-tuned only on the downstream tasks.
>
>
> **Reviewer Question:** "Can TreeTop model the associations between multiple conversation trees, especially based on longitudinal data or same user scenarios?”
>
> In principle, this is possible, however we did not experiment with any downstream tasks involving longitudinal phenomena. What we have shown is that models fine-tuned on TreeTop's primitive structural tasks are the top performers on downstream tasks involving single conversation (in some cases, SotA). Importantly, we also showed that TreeTop models also generalize to unseen primitive tasks such as wedge/interaction/conflict detection (Section 4.1, Table 2). Together, these results strongly imply that TreeTop models will generalize well to tasks with tree-structured data, including inputs with multiple trees. Out-of-the-box, TreeTop models trained on single-tree tasks could be used to output representations of all (partial) trees in the input, and then an aggregator model could be used to deliver a final inference. On the other hand, fine-tuning models to consider cross-dependencies between trees would require new primitive tasks that are promising areas for future work. We have added the following sentence to our discussion section:
>
> "Aside from multi-modality, another interesting future direction would be designing primitive tasks defined over \emph{multiple} trees, involving resolving same-user behavior and cross-dependencies, as a way to fine-tune models for more complex longitudinal effects."
>
> **Reviewer Question:** "How does TreeTop differentiate between structure- and content-related inferences when solving a reasoning task?”
>
> Thank you for the interesting question. Please note that the structure and the content are strongly dependent -- for instance, a negative reply to the top post (depth 2) should be interpreted differently than a negative reply to the reply (depth 3). We discuss this further in the paragraph at L82. Due to this dependence, our TreeTop LLM prompt takes both the structure of the conversation tree and the content on the nodes of the tree as input, and TreepTop models are fine-tuned to draw inferences from both simultaneously. The powerful transformer-based architectures of LLMs are well-suited to learn which input features to focus on, given the rest of the input.
>
> **Reviewer Question:** "How does TreeTop compare to other fine-tuning based methods on general graph-based reasoning tasks, such as edge existence prediction or node classification?”
>
> The set of 17 TreeTop primitive tasks actually includes both edge existence and node classification for tree graphs. Our edge existence task is called "are_one_hop_neighbors". We also include meta-edge existence tasks such as "are_two_hop_neighbors", "are_three_hop_neighbors", "within_subtree", and "in_triangle". The set of tasks also include multiple node classification tasks including "depth", "is_leaf", "num_children", and "num_leaf_nodes". All these tasks are detailed in Appendix C.1. Note that TreeTop tasks are defined on temporal, acyclic, directed graphs, and are meant to be analogous to those defined on undirected graphs such as those found in e.g. Guo et al 2023 and other related works discussed in Section 2.

---

> ### Author Response · Authors · 2024-11-21
>
> **Reviewer Question:** "How sensitive are the model outputs to prompt design, and is there a mechanism to automatically adjust prompts for different tasks?”
>
> To ensure that TreeTop models can translate natural language about tree structure into reasoning capability, during fine-tuning, we used multiple (2-3) prompt variants for each of the 12 primitive tasks. These prompt variants are listed in Table 10. This has multiple effects:
>
> 1. It allows TreeTop models to be robust to variations in the prompt, allowing it to effectively answer quantitative questions even when they are phrased slightly differently.
> 2. Importantly, it allows TreeTop models to generalize to unseen primitive tasks, specifically the 5 unseen tasks (see Section 4), which require new prompts on which were not part of fine-tuning.
> 3. It allows TreeTop models to be profitably trained on downstream tasks with entirely new prompts that were not used during primitive task fine-tuning.
>
> Automatic prompt generation for different tasks is out-of-scope for the TreeTop framework; we consider this to be in the scope of systems that deploy TreeTop for various downstreams. Due to the above techniques, which inject prompt robustness into TreeTop models, we expect such strategies to be effective.

---

> > ### Author Response · Authors · 2024-12-02
> > **Request for more questions and feedback**
> >
> > Thank you for your initial review. We are wondering if our response clarified matters. If so, we humbly request that you reconsider your score. If not, we are happy to answer any follow-up questions or comments. Cheers,
> >
> > Authors

---

### Official Review · Reviewer_jnJR · 2024-11-03

**Soundness:** 3
**Presentation:** 3
**Contribution:** 2
**Rating:** 6
**Confidence:** 3

**Summary:**

This paper proposes a fine-tuning method on conversation-tree related tasks.
Concretely, the conversation-tree for the LLM to understand and the corresponding tasks in the form of yes/no questions are encoded in a structured prompt and used for fine-tuning. This method is named as TreeTop. They show that TreeTop fine-tuned LLMs have better performance than their not fine-tuned counterparts across various tasks.

**Strengths:**

- a new method is proposed to encode the tree structure of the conversation-trees and is showed to be effective
- designed some structural tasks to evaluate the topological structure of the conversations trees
- the proposed fine-tuning method does not only boost the performance on fine-tuned tasks but also on unseen tasks
- extensive experiments were conducted and the ablation studies are also detailed

**Weaknesses:**

See questions

**Questions:**

- For the primitive structural tasks, why is TT not compared against the baseline where the LLM is fine-tuned with non TreeTop encoded conversations and tasks? Please point it out if I understand it incorrectly.

---

> ### Author Response · Authors · 2024-11-21
>
> We thank Reviewer jnJR for their thorough and constructive feedback. We particularly appreciate their recognition of TreeTop's key contributions: the novel tree structure encoding method, the carefully designed structural tasks, and the framework's ability to generalize to unseen tasks. We are also grateful for acknowledging the comprehensiveness of our experimental validation through extensive ablation studies.
>
> **Reviewer Question:** "For the primitive structural tasks, why is TT not compared against the baseline where the LLM is fine-tuned with non TreeTop encoded conversations and tasks? Please point it out if I understand it incorrectly."
>
> Thank you for this important question about baselines. We have also conducted ablation studies comparing TreeTop against models fine-tuned on raw conversations without our structural encoding. Such an encoding captures the content of the conversation tree, without including the structure information about parent nodes and children (referred to as the PC encoding below). We show that TreeTop outperforms Gemini tuned on non-TreeTop encoded conversations and trees, thereby strengthening the need for our TreeTop framework. We have added this encoding ablation results in Appendix H. We include a sample result below to show this improvement (ZS refers to zero-shot and FT refers to fine-tuned; all numbers are f1 scores).
>
> &nbsp;
>
> ### Controversial Post Detection
> | Encoding |   ZS	|	FT	|
> | :----------  | ------: | ------: |
> | Gemini-PC |	38.9  |		73.9 |
> | Gemini-TT |	64.7  |	 	85.0 |
>
> &nbsp;
>
> ### Fake News Detection
> | Encoding |   ZS	|	FT	|
> | :----------  | ------: | ------: |
> | Gemini-PC |	28.9 |		93.5 |
> | Gemini-TT  |	76.5  |	 	91.0 |
>
> &nbsp;
>
> ### Rumor Detection
> | Encoding |   ZS	|	FT	|
> | :----------  | ------: | ------: |
> | Gemini-PC |	49.5 |		69.7 |
> | Gemini-TT |	76.5 |	 	80.7 |
>
>
> We thank the reviewer for raising the point on encoding ablations, which has helped make our experimental validation more comprehensive. We hope you find this clarification, and our additional experiments, useful towards acceptance of our work.

---

> > ### Comment · Reviewer_jnJR · 2024-11-26
> >
> > Thanks for answering my questions addressed.
> > I am still confused by the results tables. From my understanding the Gemeni-TT result shown in Table 3 is fine-tuned with TreeTop and in Appendix H it is fine-tuned with non-Treetop encodings, why are the two the same?

---

> > > ### Author Response · Authors · 2024-11-28
> > >
> > > Table 21 in Appendix H compares 2 different encodings for the trees - namely PC and PCT (described in Table 20). PC and PCT encodings are two differing representations of the same conversation tree. The PC encoding concatenates the content of the post and comments, whereas the PCT encoding also incorporates the ‘topology’ (i.e. the parent-child relationship between two nodes). In Appendix H, we present results showing the performance of Gemini when the input uses the PC and the PCT encodings, and compare both of these to the performance of the Gemini-TT model.
> > >
> > > The first row of Table 21 shows the results for Gemini when the input is in the PC encoding. Here, each conversation is represented in the PC format, and then Gemini is fine-tuned for the downstream task (namely controversy detection, fake news detection and rumor detection). Gemini-PC shows the results of this fine-tuned Gemini model.
> > >
> > > Similarly, Gemini-PCT shows the results when each conversation tree  is encoded in the PCT format, and Gemini is fine-tuned for the downstream task.
> > >
> > > The Gemini-TT model also uses the same PCT format, but here Gemini is first fine-tuned on structural tasks, and then further fine-tuned on downstream tasks. This third row is the same result as shown in Table 3 in the earlier sections, and we show that here for completeness.
> > >
> > > These results collectively demonstrate that the Gemini model by itself is not able to leverage topology (Gemini-PC and Gemini-PCT results are almost same), and  structural fine-tuning adds a lot of value (Gemini-TT results are significantly better than both Gemini-PC and Gemini-PCT results).

---

> > > > ### Author Response · Authors · 2024-12-02
> > > > **Request for more questions and feedback**
> > > >
> > > > Thank you for your initial review and your reply to our rebuttal. We are wondering if our further response clarified matters. If so, we humbly request that you reconsider your score. If not, we are happy to answer any follow-up questions or comments. Cheers,
> > > >
> > > > Authors

---

### Official Review · Reviewer_8zwd · 2024-11-04

**Soundness:** 2
**Presentation:** 1
**Contribution:** 2
**Rating:** 3
**Confidence:** 4

**Summary:**

This paper introduces TreeTop, a fine-tuning framework designed to enhance the performance of LLMs in understanding and reasoning over conversation trees—structured, multi-party discussions typically found on social media. The framework provides a collection of 17 structural tasks, aiming to improve LLMs' capacity to process both content and the structure of conversation trees. Experimental results show that LLMs fine-tuned with TreeTop outperform baseline models, including state-of-the-art GNNs, on multiple social media inference tasks, such as controversy and rumor detection.

**Strengths:**

- The paper tackles a novel challenge by focusing on conversation trees, which have unique characteristics such as directed, acyclic, and temporal structures. This addresses an important gap in LLM capabilities.
- TreeTop shows potential across a range of tasks (e.g., controversy detection, rumor detection) relevant to social media, making the framework broadly useful.

**Weaknesses:**

- Some tasks are too easy to be meaningful. This work is proposed as a benchmark for LLMs and what's the point if a LLM can already achieve 100% accuracy?
- Most datasets used in this study are derived from Reddit, which may limit the generalizability of TreeTop’s effectiveness on other social media platforms or discussion types.
- The paper briefly touches on potential issues such as over-moderation or promoting echo chambers but does not provide a detailed discussion on these risks or propose safeguards.
- The paper only includes results from one model (Gemini), which is bluntly telling the authors work for a certain company. Also not including other models fail to set meaningful baselines.
- Format: This paper uses a wrong font.

**Questions:**

See "Weaknesses."

---

> ### Author Response · Authors · 2024-11-21
>
> We sincerely thank Reviewer 8zwd for their time spent reviewing our paper. We also thank them for recognizing the novelty of our work in tackling challenges related to understanding conversation trees. We address the concerns raised as follows:
>
> **Weakness 1:**
>
> **Reviewer Comment:** "Some tasks are too easy to be meaningful. This work is proposed as a benchmark for LLMs and what's the point if a LLM can already achieve 100% accuracy?"
>
> We appreciate the fair question, however, our experiments actually show that LLMs *cannot* achieve 100% accuracy on any task from the TreeTop framework without directly fine-tuning on that task. For instance, as shown in Table 2, Gemini's zero-shot and few-shot accuracies on the 12 TreeTop tasks chosen for fine-tuning largely fall in the 40%-70% range. Only after fine-tuning on those 12 tasks do accuracies increase to 93%-100%. Moreover, post-fine-tuning accuracy on the 5 tasks *held-out* during fine-tuning only increases to 63%-86%, showing that these tasks are in fact quite non-trivial.
> These results validate the core ideas of TreeTop: out-of-the-box, LLMs are not good at reasoning over structured data (see also Table 1). So, we introduce a suite of general, downstream-agnostic reasoning tasks on which they can be profitably fine-tuned (note that we do not propose this work as a benchmark). We show that fine-tuning on these tasks greatly increases the potential of models for out-of-distribution ("unseen") tasks and downstream applications.
>
> **Weakness 2:**
>
> **Reviewer Comment:** "Most datasets used in this study are derived from Reddit, which may limit the generalizability of TreeTop's effectiveness on other social media platforms or discussion types."
>
> We respectfully disagree with the concern about dataset generalizability, as our experimental results demonstrate TreeTop's robust cross-platform effectiveness. The framework captures invariant structural properties of conversation trees that are fundamentally consistent across social media platforms. This is validated by our results on the PHEME9 Twitter dataset, where TreeTop-enhanced models achieve 87.1% accuracy - significantly outperforming prior approaches like PHAROS (75.9%) and NRA MOS-GAT (78.4%). The platform-agnostic nature of TreeTop is supported by our analysis of tree topologies, which shows that the framework learns structural patterns rather than platform-specific features.
> While it is true that most of our datasets come from Reddit, this is largely due to practical constraints: publicly available datasets with complete post-comment trees are scarce, especially from platforms like Twitter, which now require paid API access. Despite these data availability challenges, our transfer learning experiments demonstrate that models trained on Reddit data maintain strong performance on Twitter data without platform-specific adjustments (as shown by our results on the Pheme9 dataset). This effectiveness stems from TreeTop's focus on fundamental conversation tree properties that are consistent across platforms: hierarchical parent-child relationships, branching reply structures, and temporal dynamics. The topology of these trees follows measurable patterns in depth distribution and interaction dynamics across Reddit and Twitter. The empirical results establish that the TreeTop framework generalizes effectively across different social media platforms.
>
> **Weakness 3:**
>
> **Reviewer Comment:** "The paper briefly touches on potential issues such as over-moderation or promoting echo chambers but does not provide a detailed discussion on these risks or propose safeguards."
> We appreciate the opportunity to provide more detail. We have added the following sentences to Section 5. These sentences replace the sentence starting at L502 ("Second, we note that...") and continue until the end of the paragraph:
> "We also note that models such as those fine-tuned on TreeTop are often used for content-moderation (Roberts, 2017; Gillespie, 2020). A serious risk for such models centers around their potential misuse for content over-moderation, thereby promoting echo-chambers and insularity of thought (Kumar et al., 2024).  Practitioners and platform designers should ensure that auto-moderation models trained with TreeTop (or any similar framework) are deployed responsibly, such as preferring “soft” moderation tags (shown in Martel & Rand (2023) to be effective) over bans/takedowns, providing definitional clarity to moderation policies, and offering transparency into moderation decisions, such as open-sourcing logs (Macdonald & Vaughan, 2024; Singhal et al., 2023).  Additionally, our experiments show that TreeTop models are performant at early-detection (68.9% accuracy at 4 hours), which can enable effective human-in-the-loop moderation systems to be designed (Lai et al., 2022)."

---

> ### Author Response · Authors · 2024-11-21
>
> **Weakness 4:**
>
> **Reviewer Comment:** "The paper only includes results from one model (Gemini), which is bluntly telling the authors work for a certain company. Also not including other models fail to set meaningful baselines."
>
> We appreciate the reviewer’s feedback, and yet would like to point towards the extensive experiments done using different classes of models – Gemini, PaLM-Bison and Gemma, while also considering ablation studies with Phi Mini and Mistral 7B (See Table 1, Page 6).  The results show consistent improvement across all families - for instance, on structural tasks, Gemma-TT achieves a significant improvement in accuracy (90-100%) on most fine-tuning tasks and significant gains on unseen tasks (e.g., 80.8% on user interaction detection as compared with 22.1% accuracy prior to fine-tuning). Similarly, PaLM-TT shows strong results with 90-99% accuracy across most structural fine-tuning tasks on the hold-out set, validating that TreeTop's benefits generalize across model architectures.
>
> For baselines, we provide extensive comparisons with other comparable approaches. For example, we compare against both traditional GNN architectures (GCN, GAT, GraphSage) and specialized models like DFE-GCN for controversy detection (76.6% accuracy vs TreeTop's 84.6%), PHAROS for rumor detection (75.9% vs 87.1%), and domain-specific approaches for winning argument detection. These comparisons demonstrate TreeTop's effectiveness is not model-specific but rather stems from its fundamental approach to structural learning.
>
> Additionally, we validate our results through multiple ablation studies across models and datasets (detailed in Appendices G and I), ensuring that our findings generalize across different conditions. These empirical findings provide strong evidence for our framework’s general applicability across baselines.
>
> **Weakness 5:**
>
> **Reviewer Comment:** “Format: This paper uses a wrong font.”
> We thank the reviewer for pointing this out. While we were indeed using the ‘iclr2025_conference’ template available [here](https://github.com/ICLR/Master-Template/raw/master/iclr2025.zip), our inclusion of package ‘lmodern’ did conflict with a font. We have removed this offending package and uploaded a revision consistent with the styling requirements of ICLR.
>
> We thank Reviewer 8zwd for their valuable feedback. We hope our comments and the revisions satisfactorily address your concerns. We request the reviewer to reconsider our work in light of these responses. If they think we have sufficiently addressed their concerns, we kindly request them to consider raising their score.

---

> > ### Author Response · Authors · 2024-12-02
> > **Request for more questions and feedback**
> >
> > Thank you for your initial review. We are wondering if our response clarified matters. If so, we humbly request that you reconsider your score. If not, we are happy to answer any follow-up questions or comments. Cheers,
> >
> > Authors

---

### Author Response · Authors · 2024-11-21

We sincerely thank all the reviewers for their thoughtful comments and valuable feedback on our work. We are grateful that Reviewer 8zwd recognized the novelty of our approach in addressing the challenges of understanding conversation trees. We appreciate Reviewer jnJR and Reviewer iBfG for acknowledging the comprehensiveness of our structural tasks and the effectiveness of our proposed method. We also thank Reviewer 5ohp for noting the potential benefits of our new conversation tree encoding method to the field of multi-turn, multi-party conversations.

Multiple reviewers asked thoughtful questions about the purpose and impact of TreeTop. As a general response, we would like to point out that TreeTop is a fine-tuning framework for LLMs (not a benchmark), consisting of 17 novel reasoning tasks on conversation trees on which pretrained LLMs can be fine-tuned. We hypothesize that fine-tuning on these tasks will allow LLMs to generalize to a variety of downstream tasks defined on conversation trees. To demonstrate this promise of TreeTop, we fine-tune multiple LLMs on 12 of these tasks, and then analyze the performance of the models fine-tuned with TreeTop vs zero/few-shot performance. We find that:

1) Without TreeTop fine-tuning, LLMs perform poorly on all 17 primitive reasoning tasks that we introduce, showing that the tasks are non-trivial for out-of-the-box LLMs.

2) On the 12 TreeTop tasks included for fine-tuning, LLM performance rises from ~40-60% to near-perfect (93-100%).

3) On the 5 TreeTop tasks held out for evaluation (not included in fine-tuning), LLM performance rises by 10-20%.

4) Across-the-board, on a variety of downstream tasks from multiple platforms, LLMs fine-tuned with TreeTop outperform all competitors (including non-LLM SotA and LLMs not fine-tuned with TreeTop). This result holds across a large variety of settings that we have carefully tested with controlled experiments and ablations.

5) An additional notable result of the TreeTop framework is its capacity for early detection. This early warning capability enables effective human-in-the-loop moderation while preventing over-automation, enabling significant real-world impact.

Based on the reviewers' responses, we have made several changes to the draft to improve the clarity. Specifically, we have reworked parts of Section 3 and Section 4 (particularly S4 intro and S4.1) to clarify the message above. We have highlighted the details about finetuning experiments at the outset of Section 4. We have expanded the discussion section (Section 5) and now address multiple points raised by the reviewers like platform generalization beyond Reddit, and content moderation implications. We also added an entirely new Appendix H providing new ablation results using an alternate encoding of the conversation tree. We would greatly appreciate the reviewers' consideration of these revisions.

## Addressing General Questions:

**Baselines and Comparisons with Other Models (Reviewer 1 and Reviewer 3):**

**Q1 (Reviewer 8zwd):** "The paper only includes results from one model (Gemini)... Also not including other models fail to set meaningful baselines."

**Q2 (Reviewer iBfG):** "In terms of comparisons, only traditional GNNs are used as the main baselines, lacking comparisons with more recent graph-based reasoning methods."

**Response:** We thank the reviewers for raising these important points about model comparisons and baselines. Our evaluation includes experiments across multiple model families and baselines. The results demonstrate TreeTop's effectiveness across three different LLM architectures: Gemma-2B (open source) achieving 90-100% accuracy on the holdout set of structural fine-tuning tasks, PaLM-Bison showing 90-99% accuracy, and Gemini-Pro reaching 93-100% accuracy (see Table 2, and Appendix G). These results show that TreeTop's benefits generalize across different architectures and are not specific to any single model family.

Beyond LLM comparisons, we evaluate against both traditional GNN architectures (GCN, GAT, GraphSage) and recent specialized approaches. On controversy detection, as shown in Table 3, TreeTop (84.6%) significantly outperforms the recent DFE-GCN approach (76.6%). For rumor detection, as shown in Table 16, TreeTop achieves 87.1% accuracy, substantially exceeding both PHAROS (75.9%) and NRA MOS-GAT (78.4%). The consistent improvement across both model families and baseline comparisons demonstrates that TreeTop's effectiveness stems from its fundamental approach to structural learning rather than from any specific model architecture. Our ablation studies, detailed in Appendices E, G, H, I further validate these findings.

---

### Author Response · Authors · 2024-11-25

We thank the reviewers for their time. We wanted to encourage the reviewers to reconsider the ratings. We have tried to provide detailed responses to the reviews along with the experimentations whenever required. We have also updated the paper pdf accordingly. We would also be happy to respond to any reviewer comments promptly given the closeness of the end of the discussion period.

---

### Author Response · Authors · 2024-11-25

We thank the reviewers for their time. We wanted to encourage the reviewers to reconsider the ratings. We have tried to provide detailed responses to the reviews along with the experimentations whenever required. We have also updated the paper pdf accordingly. We would also be happy to respond to any reviewer comments promptly given the closeness of the end of the discussion period.

---

### Meta-Review · Area_Chair_89pY · 2024-12-20

**Metareview:**

This paper introduces TreeTop, a framework designed to enhance the capabilities of LLMs in processing conversation trees, particularly those with directed, acyclic, and temporal structures, tackling a significant gap in existing LLM methodologies. The framework shows potential in various tasks relevant to social media, such as controversy and rumor detection.

However, as pointed out by the reviewers, there are a few concerns: some tasks appear too simplistic, achieving near-perfect accuracy, which may not serve as meaningful benchmarks; the generalizability of the results is questionable due to the predominant use of data derived from Reddit; the novelty of the paper; and some others. Even though the authors tried to address these concerns during the rebuttal, the reviewers were not fully satisfied.

**Additional Comments On Reviewer Discussion:**

Nil.

---

### Decision · Program_Chairs · 2025-01-22

Reject